# Transcriptomic analysis to identify genes associated with selective hippocampal vulnerability in Alzheimer's disease

Angela M. Crist [1], Kelly M. Hinkle[1], Xue Wang[2], Christina M. Moloney [1], Billie J. Matchett [1], Sydney A. Labuzan [1], Isabelle Frankenhauser [1,3], Nkem O. Azu[1], Amanda M. Liesinger[1], Elizabeth R. Lesser [2], Daniel J. Serie[2], Zachary S. Quicksall[2], Tulsi A. Patel[1], Troy P. Carnwath[1], Michael DeTure[1], Xiaojia Tang[4], Ronald C. Petersen[5], Ranjan Duara[6], Neill R. Graff-Radford[7], Mariet Allen[1], Minerva M. Carrasquillo [1], Hu Li [8], Owen A. Ross [1], Nilüfer Ertekin-Taner[1,7], Dennis W. Dickson [1], Yan W. Asmann[2], Rickey E. Carter [2] & Melissa E. Murray [1✉]

Selective vulnerability of different brain regions is seen in many neurodegenerative disorders. The hippocampus and cortex are selectively vulnerable in Alzheimer's disease (AD), however the degree of involvement of the different brain regions differs among patients. We classified corticolimbic patterns of neurofibrillary tangles in postmortem tissue to capture extreme and representative phenotypes. We combined bulk RNA sequencing with digital pathology to examine hippocampal vulnerability in AD. We identified hippocampal gene expression changes associated with hippocampal vulnerability and used machine learning to identify genes that were associated with AD neuropathology, including *SERPINA5*, *RYBP*, *SLC38A2*, *FEM1B*, and *PYDC1*. Further histologic and biochemical analyses suggested SERPINA5 expression is associated with tau expression in the brain. Our study highlights the importance of embracing heterogeneity of the human brain in disease to identify disease-relevant gene expression.

[1] Department of Neuroscience, Mayo Clinic, Jacksonville, FL, USA. [2] Department of Health Sciences Research, Mayo Clinic, Jacksonville, FL, USA. [3] Paracelsus Medical Private University, Salzburg, Austria. [4] Department of Health Sciences Research, Mayo Clinic, Rochester, MN, USA. [5] Department of Neurology, Mayo Clinic, Rochester, MN, USA. [6] Wien Center for Alzheimer's Disease and Memory Disorders, Mount Sinai Medical Center, Miami Beach, FL, USA. [7] Department of Neurology, Mayo Clinic, Jacksonville, FL, USA. [8] Department of Molecular Pharmacology and Experimental Therapeutics, Mayo Clinic, Rochester, MN, USA. ✉email: murray.melissa@mayo.edu

Alzheimer's disease (AD) is a multi-factorial neurodegenerative disorder characterized under the microscope by the abnormal accumulation of two misfolded proteins, tau and amyloid-β (Aβ)[1]. While both are hallmarks of the disease, they accumulate in very different fashions. Aβ accumulation is the result of enzymatic cleavage of the amyloid precursor protein[2,3], whereas abnormal tau accumulation occurs following post-translational modification that disrupts its utility in microtubule stabilization[4,5]. Aβ plaque deposits accumulate in the extracellular space of isocortical convexities before descending into the limbic system, diencephalon, brainstem, and lastly cerebellum according to Thal amyloid phase[6]. In contrast, abnormally folded tau accumulates intracellularly as neurofibrillary tangles in subcortical nuclei and entorhinal cortex before ascending to limbic structures, association cortices, and lastly primary cortices according to Braak tangle stage[7]. Thus, two partners in the devastation of the AD brain with divergent topographic origins intersect in the limbic system, and more specifically the hippocampus, before continuing on their unique neuroanatomic journey.

Studies demonstrate Aβ plaque pathology as predictable and rarely deviating from the well-established Thal amyloid phase[8,9]. However, we identified three AD subtypes using an objective mathematical algorithm to assess corticolimbic patterns of neurofibrillary tangle pathology: hippocampal sparing AD, typical AD, and limbic predominant AD[10–13]. Typical AD brains were representative of the expected patterns of hippocampal and cortical involvement as outlined by Braak tangle stage. In contrast, we discovered two extremes that exist outside of the Braakian-concept of neurofibrillary tangle patterns. Hippocampal sparing AD cases demonstrate selective resilience of the hippocampus relative to severely involved association cortices, whereas limbic predominant AD cases demonstrate inundation of the hippocampus relative to mildly involved association cortices (Fig. 1a). Further examination of their clinical relevance revealed a constellation of demographic and clinical differences that were monotonically directed among AD subtypes, including striking differences in age at onset, sex, and APOE ε4 status[10–13]. Given the neuropathologic and clinical differences observed among AD subtypes, we sought to test the hypothesis that the objective classification of disease spectrum could be leveraged to uncover transcriptomic changes that underlie selective vulnerability of the hippocampus in AD. Herein, we outline our translational neuropathology approach for gene prioritization that, coupled with machine learning, can be used to identify biologically relevant gene expression changes throughout a disease spectrum.

## Results

### Bulk transcriptional profiling of hippocampal vulnerability in AD.
Postmortem human brain samples of neuropathologically diagnosed AD cases lacking co-existing neuropathologies and nondemented controls were derived from the Mayo Clinic Brain Bank (Fig. 1a and Supplementary Fig. 1). We obtained bulk transcriptomic data from the hippocampi of 40 AD cases and 15 controls (Fig. 1b) before employing a five-step multi-disciplinary approach toward gene prioritization with the overall goal of selecting disease-relevant, protein-coding genes (Fig. 2). Our first step was to select genes from the literature associated with clinical expression of AD. We selected 25 genes located at loci identified in genome-wide association studies (GWAS) of late-onset AD[14–19], three genes associated with early-onset AD[20,21], and three additional genes identified to modify the phenotype of AD[22–24] (Fig. 2 [Step 1]). Our second step was to utilize our prior knowledge of disease spectrum classified by the AD subtype algorithm[13] (Fig. 2 [Step 2]). Therefore, we divided our RNA sequencing (RNA-Seq) cohort into two groups: representative phenotype (typical AD compared to control) and extreme phenotype (limbic predominant AD compared to hippocampal sparing AD) (Fig. 2). Given the constellation of clinicopathologic features found to differ among AD subtypes, we did not adjust RNA-Seq data by age, sex, or APOE ε4 status. Differential expression analysis of the representative phenotype identified 1407 genes, among 19,720 genes that were expressed (Fig. 2 [Step 2a]). Of the 1407 differentially expressed genes in the representative phenotype, 141 genes met bioinformatic prioritization criteria and remained for further analyses of the representative phenotype (Fig. 2 [Step 2b]). Differential expression analysis of the extreme phenotype identified 349 genes, among 19,747 genes that were expressed (Fig. 2 [Step 2a]). Following bioinformatic prioritization, 73 genes remained for further analyses of the extreme phenotype (Fig. 2 [Step 2b]). Of the 186 unique genes prioritized in Step 2b, only 28 (15%) overlapped between the representative phenotype and extreme phenotype.

In both the representative phenotype and extreme phenotype, a majority of the genes were downregulated (Fig. 3a, b). There were 106/141 (75%) downregulated genes identified in the representative phenotype and 60/73 (82%) downregulated genes identified in the extreme phenotype. No upregulated genes were observed to overlap (Fig. 3b). These results suggest that within the spectrum of a disease, the pattern of upregulation or downregulation observed between disease versus controls may be recapitulated in the extremes of the disease state. In the current study, the degree of log_2 fold change (log2FC) observed in the downregulated genes was nearly twice that observed in the upregulated genes (Fig. 3c).

Differentially expressed genes in isolation may not be representative of their own importance to the phenotype, but they could instead reflect the indirect relationship with a gene network[25]. Thus, our third step was to identify differentially expressed genes from Step 2a that were enriched in pre-built process networks using a curated database created by MetaCore™ to model main cellular processes (Fig. 2 [Step 3]). All genes from the top process networks, that were selected for the representative phenotype and separately for the extreme phenotype, were then included as candidate genes. The representative phenotype was highly enriched for genes associated with cell growth and proliferation, whereas the extreme phenotype was highly enriched for genes associated with inflammation and immune response (Fig. 3d, e). A total of 111 genes were selected from the top ten process networks identified in the representative phenotype and 34 genes were selected from those identified in the extreme phenotype (Fig. 2 [Step 3]).

### Translational neuropathology approach.
Our overall goal was to identify biologically meaningful gene expression changes that relate to AD neuropathology and cognition. Following Steps 1–3, we narrowed down our gene set to a total of 339 unique genes to be reviewed in our fourth step toward final gene prioritization (Fig. 2 [Step 4]). The disease spectrum was separated back into four distinct groups to enforce monotonic directionality of gene expression changes in an expected direction of upregulation (control < hippocampal sparing AD < typical AD < limbic predominant AD) as exampled by SIRT1 or downregulation (limbic predominant AD < typical AD < hippocampal sparing AD < control) as exampled by PSEN2 (Fig. 3f, g and Supplementary Fig. 2). Of the 182 genes that showed group-wise differences in gene expression ($p < 0.05$), 79 were monotonically directed.

Next, we investigated the relationship of the remaining 79 genes with neuropathologic burden of tau and Aβ across serial hippocampal sections for each AD case and control using digital pathology to quantify disease burden[10,11,13,26–28]. Tau

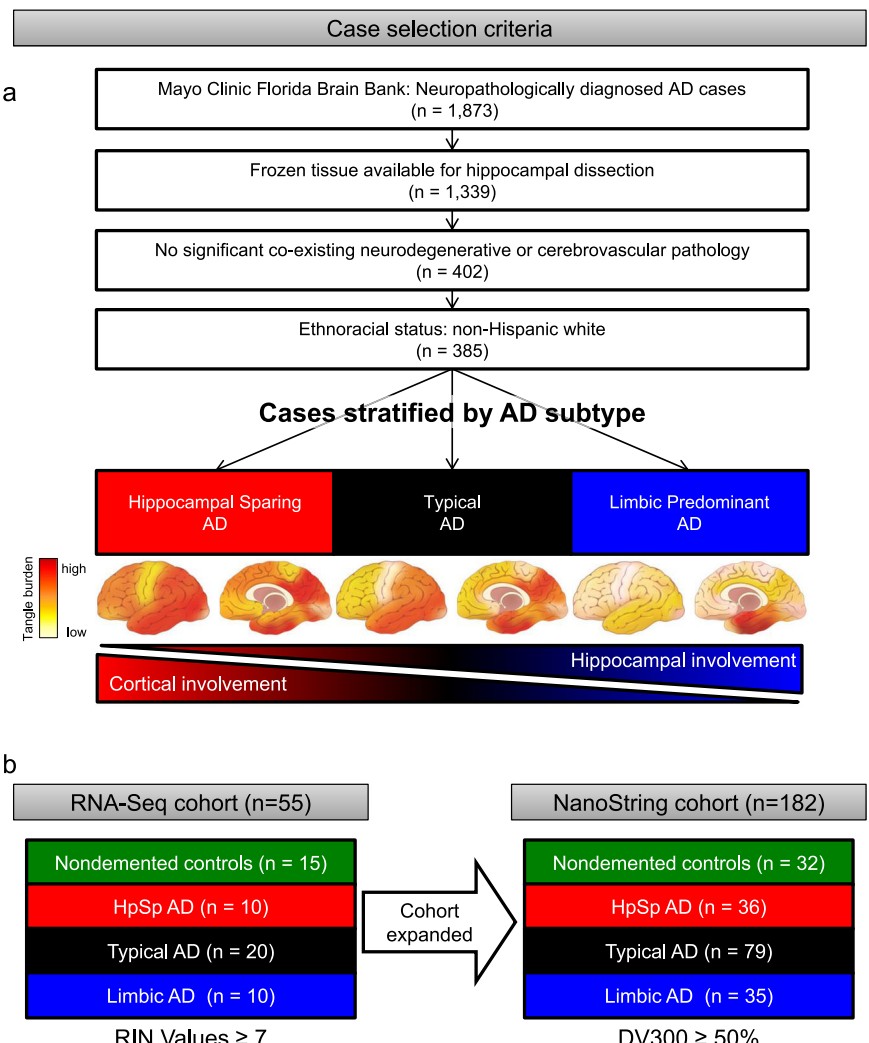

**Fig. 1 Case selection criteria and neuropathologic subtyping of AD. a** Flowchart describing selection criteria of AD cases to obtain frozen hippocampal tissue used to perform RNA-Seq and NanoString studies. AD cases were subtyped using a validated algorithm based on topographic distribution of hippocampal and cortical neurofibrillary tangles[13]. Brain cartoons illustrate corticolimbic patterns of neurofibrillary tangle burden throughout the brain in each AD subtype, where red indicates a high tangle burden and yellow/tan indicates a low tangle burden. **b** Overview of sample size for RNA-Seq and NanoString cohorts organized by AD subtype and control classification. Note: Cohort expansion for NanoString included all RNA-Seq cohort samples. AD, Alzheimer's disease; DV300, distribution value over 300 base pairs; HpSp, hippocampal sparing; Limbic, limbic predominant; NFT, neurofibrillary tangle; RIN, RNA integrity number; RNA-Seq, RNA sequencing.

accumulation inside the neuron does not occur in isolation as a static entity, but instead it accumulates along the lifespan of a neurofibrillary tangle[29,30]. To quantitatively capture the dynamic range of neurofibrillary tangle maturity, we used an early marker of maturity (CP13, Fig. 3h)[31] and an advanced marker of maturity (Ab39, Fig. 3i)[32]. Data are presented in Supplementary Fig. 1. The early marker of tangle maturity in the hippocampus was not found to differ among AD subtypes (AD specific $p$ = 0.11); however, the advanced marker of tangle maturity was found to differ (AD specific $p$ = 0.018). This was not surprising given the abundance of pretangle and neuritic pathology recognized by the CP13 antibody[31], in comparison to the Ab39 antibody which recognizes advanced tangle pathology (i.e., mature tangles and extracellular tangles) that better reflects tau-induced neurodegeneration. Aβ burden was quantified with a pan-Aβ marker (33.1.1, Fig. 3j) to ensure the capture of all plaque types. We did not observe a difference in Aβ burden among AD subtypes (AD specific $p$ = 0.94). To complement these local measures of pathology in the hippocampus, we additionally investigated the relationship of the prioritized genes with Braak

tangle stage[7] and Thal amyloid phase[6] as global measures of AD pathology (Supplementary Fig. 3).

The remaining genes to be evaluated in Step 4 were highly enriched for protein-coding genes (72/79 [91%]), which we focused on for evaluation of neuropathologic association. Of the 72 protein-coding genes, 44 robustly associated with a measure of tau or Aβ pathology (Supplementary Fig. 3). We did not identify any genes in this final set that associated with the pan-Aβ marker (33.1.1), but 23/44 (52%) were associated with Thal amyloid phase. A majority of the genes were associated with tau pathology: advanced marker of tangle maturity (Ab39, 35/44 [80%]) or Braak tangle stage (13/44 [30%]). Gene selection from Steps 1–4 are visually represented using volcano plots of log2FC versus significance for differential gene expression in the hippocampus (Fig. 3k, l).

**Machine learning to identify associations with AD neuropathology.** To examine biological significance of the 44 prioritized genes, we expanded our gene expression studies from 55

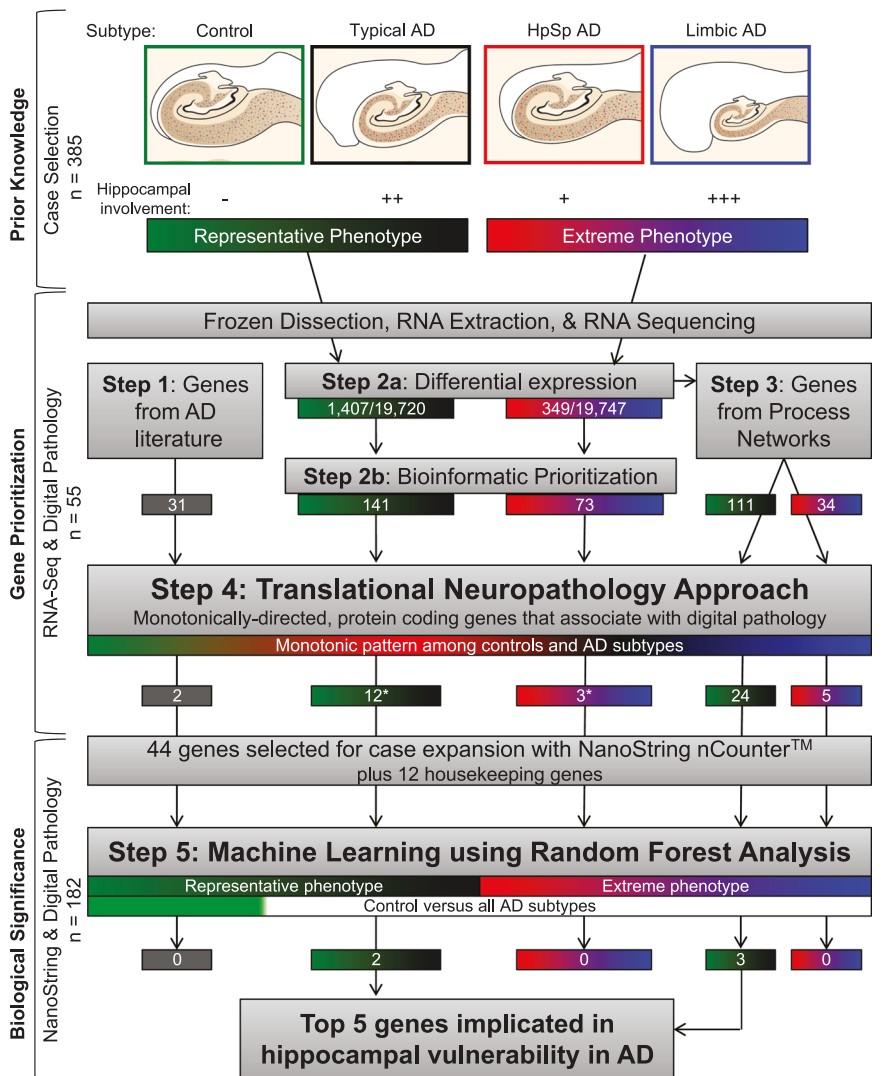

**Fig. 2 Workflow depicting multi-disciplinary method to identify genes involved in hippocampal vulnerability in AD.** Controls and typical AD cases were grouped into the representative phenotype (green-black), while hippocampal sparing AD and limbic predominant AD were grouped into the extreme phenotype (red-blue). Frozen human hippocampal tissue was first dissected, followed by RNA extraction, purification, and lastly prepped for RNA-Seq. A multi-step approach was implemented for gene prioritization, which began with genes nominated from the AD literature (Step 1). Next, we examined differentially expressed genes derived from the representative phenotype and extreme phenotype (Step 2a) before further bioinformatic prioritization (Step 2b). The differentially expressed genes identified in Step 2a were further prioritized based upon process network interactions (Step 3). We then combined the 339 unique genes from Steps 1–3 to apply a translational neuropathology approach (Step 4) whereby protein-coding genes needed to exhibit monotonic directionality and associate with local neuropathologic markers (tau and amyloid-β) or global measures of AD pathology (Braak stage and Thal phase). Once genes were prioritized, biological significance was investigated first by expanding the RNA-Seq cohort to a total of $n = 182$ in NanoString analyses. Machine learning was applied to NanoString data using the random forest algorithm (Step 5). Random forest was applied to the phenotypic groups, in addition to controls versus all AD subtypes (depicted by green versus white bar) to identify the top five genes predictive of hippocampal vulnerability in AD as a whole. This enabled us to utilize objective classification of disease spectrum to uncover transcriptomic changes that underlie selective vulnerability of the hippocampus in AD. AD, Alzheimer's disease; DE, differential expression; HpSp, Hippocampal sparing; Limbic, limbic predominant; RNA-Seq, RNA sequencing. Note: Numbers in boxes represent total gene number from each part of workflow. Steps 1–3 culminated in 339 unique genes, of which 19 overlapped in one or more of the steps. *In Step 4, one gene appeared in both groups.

hippocampi evaluated using RNA-Seq (Supplementary Fig. 1) to 190 hippocampi to be evaluated using the NanoString nCounter platform (Supplementary Fig. 4). The NanoString platform provided an affordable approach to multiplexing samples by allowing the user to design a custom codeset (i.e. gene list)[33]. Following $\log_2$ transformation of normalized NanoString gene expression, we identified seven AD cases that would require a 2.5-fold adjustment with respect to housekeeping gene correction (Supplementary Fig. 5). An additional case was excluded due to low gene expression counts overall. These eight cases were excluded

from further analyses, resulting in a final NanoString cohort of 182 AD cases and controls (Fig. 1b). As expected, gene expression levels derived from RNA-Seq and NanoString platforms showed a high degree of correlation (Supplementary Fig. 6).

Machine learning was then employed in our fifth step to assess the top genes predictive of hippocampal vulnerability in AD (Fig. 2 [Step 5]). Our first two in silico experiments examined the top genes in the representative phenotype and in the extreme phenotype. Specifically, we employed a classification and regression method known as random forest to identify the top genes

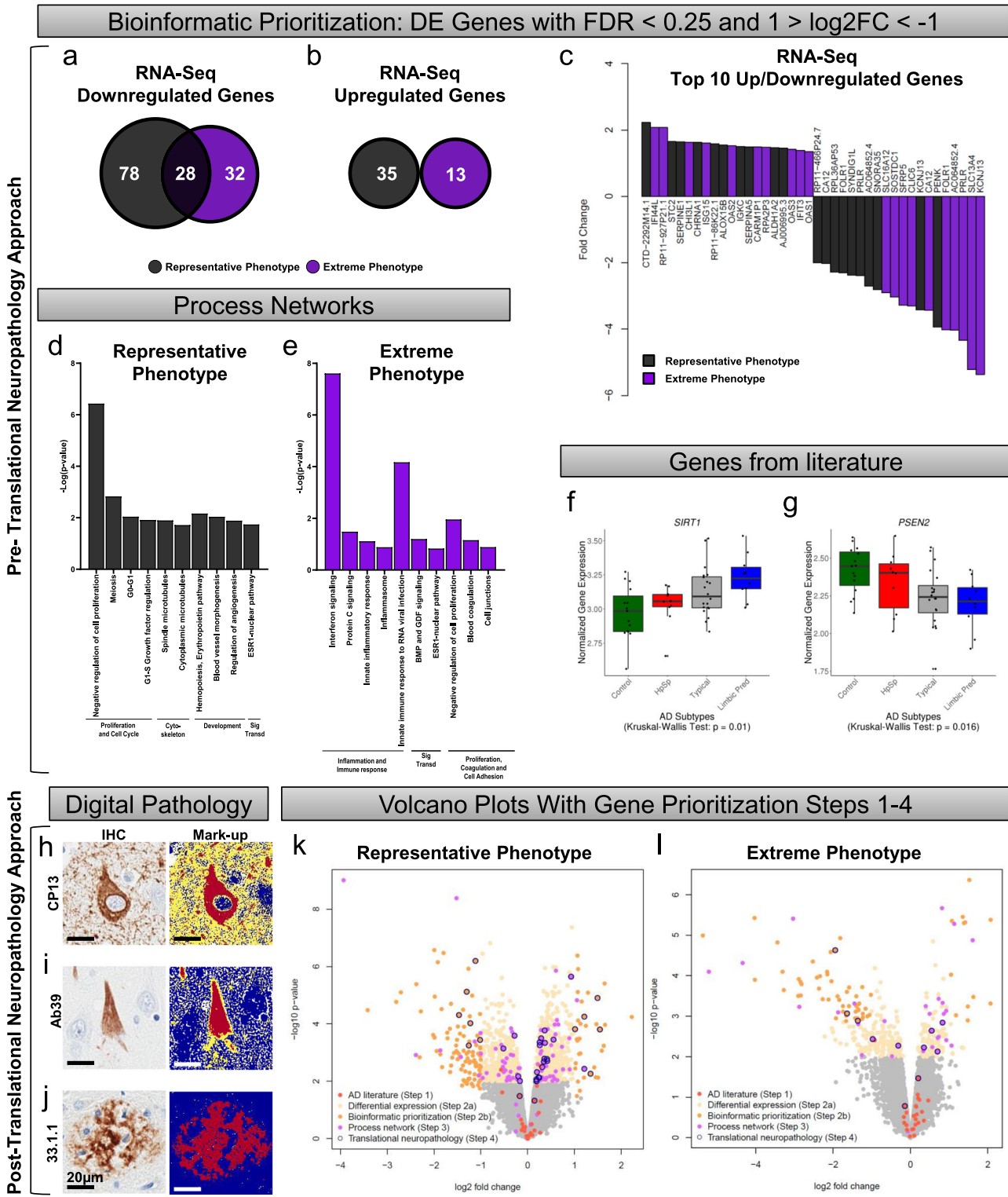

predictive for the representative phenotype and which were predictive for the extreme phenotype (see example in Supplementary Fig. 7). Random forest uses an ensemble of simple decision trees to classify data[34]. To facilitate prioritization and feasibility of evaluating top genes, mean minimum depth was used to rank the genes and a cutoff at the top five was applied (Supplementary Results 1). Out of the top five genes predictive when the representative phenotype and the extreme phenotype were separately analyzed, only *SERPINA5* overlapped between phenotypes (Supplementary Fig. 8).

The aforementioned in silico experiments provide insight into the molecular differences among AD subtypes. To maximize applicability to selective hippocampal vulnerability in AD as a whole, we combined all AD subtypes together and generated a predictive model for subsequent analysis (Fig. 4). Through inspection of the variable importance summaries (Fig. 4a, b), the following genes were considered for best discrimination of AD from control: *SERPINA5*, *RYBP*, *SLC38A2*, *FEM1B*, and *PYDC1*. A logistic regression model was fit to the data to estimate the ability for our top five genes, in concert with age and sex, to

**Fig. 3 Transcriptomic analysis reveals an abundance of downregulated genes across the disease spectrum of hippocampal vulnerability in AD. a, b** Venn diagram depicting the number of genes that were downregulated (**a**) or upregulated (**b**) in the representative phenotype (black) or extreme phenotype (purple). There was no overlap in upregulated genes. **c** Waterfall plot of the top ten upregulated or downregulated genes from each phenotype. **d, e** Process networks in the representative phenotype (black [**d**]) and extreme phenotype (purple [**e**]) reveal non-overlapping functions. **f, g** Examples of genes selected from the literature that exhibit monotonic directionality. *SIRT1* is monotonically upregulated (**f**) while *PSEN2* is monotonically downregulated (statistical values can be found in Supplementary Fig. 2) (**g**). **h–j** Digital pathology examples of immunohistochemistry and burden analysis markup using **h** CP13, **i** Ab39, and **j** 33.1.1 antibodies in CA1 subsector of the hippocampus of typical AD brains. Digital pathology measures were performed in those with sufficient tissue for serial immunohistochemical staining ($n = 54/55$ for RNA-Seq cohort and $n = 170/182$ for NanoString cohort). **k, l** Volcano plots showing fold change versus significance in the representative phenotype (**k**) and the extreme phenotype (**l**). Volcano plots reveal number of differentially expressed genes out of the total number of genes and subsequently show each step of our prioritization process: Step 1, literature-based genes (dark orange); Step 2a, differentially expressed genes (tan); Step 2b, bioinformatic prioritization of differentially expressed genes (orange); Step 3, process networks (magenta); and Step 4, translational neuropathology approach to identify NanoString validation genes (blue outline). AD, Alzheimer's disease; IHC, immunohistochemistry; RNA-Seq, RNA sequencing; Sig Transd, signal transduction. Note: Box plots in **f, g** are derived from $n = 55$ independent samples. Box plots are displayed at the 25th and 75th percentiles with the median line. Whiskers are drawn to the largest and smallest values that are within 1.5 times the interquartile range from the upper or lower quartile. For observations larger or smaller than this distance, they are shown as observations outside the whiskers.

discriminate between neuropathologically diagnosed AD cases and controls. The model provided strong discrimination with an area under the curve of 92.6% (95% confidence interval [CI]: 87.9–97.2%; Fig. 4c). To represent the model visually (Fig. 4d), a nomogram was created[35,36]. The utility of the nomogram enables each individual's demographic and gene expression data to be converted to a point-value (Fig. 4d, top), which is summarized into a total number of points that directly relates to a predicted value (Fig. 4d, bottom). The higher the predicted value, the more likely the data are representative of a neuropathologically diagnosed AD case. The nomogram visually provides a quantitative contribution of key factors like sex and age compared to gene expression values from the hippocampus (Supplementary Fig. 9). Examples of the utility of the nomogram in a nondemented control and an AD case are provided in Supplementary Fig. 10. The summarized predictive value was visually distinct between Braak tangle stages 0–III versus IV–VI and Thal amyloid phases 0–2 versus 3–5 when plotted (Fig. 4e, f). On a clinical level, we observed a non-linear relationship between the last Mini Mental State Examination (MMSE[37]) score and nomogram predictive value, with higher MMSE scores (i.e. better cognition) corresponding to lower predicted value (Fig. 4g).

To examine the relationship between gene expression changes in each of the top five genes with local measures of neuropathology in the hippocampus, we performed a series of multiple linear regression models. Gene expression levels along with selected covariates (age, sex, and *APOE* ε4) were plotted against the following measures of neuropathologic burden: early marker of tangle maturity (CP13), advanced marker of tangle maturity (Ab39), pan-Aβ pathology (33.1.1), and cell-type specific markers to adjust for contribution of microgliosis (CD68), astrogliosis (GFAP), and endothelial burden (CD34). A neuronal marker was not included in the models to avoid regressing out genes relevant to tangle-bearing neurons. A radar plot visually displays the unique relationship between each of the prioritized genes and their degree of association with markers of neuropathologic burden (Fig. 4h). Each of our top five genes correlated with different aspects of AD pathology (results summarized in Supplementary Fig. 11).

**Serine protease inhibitor SERPINA5 identified as tau interactor.** To validate the utility of our translational neuropathology approach to identify disease-relevant gene expression changes, we assessed the biological relevance of our top gene *SERPINA5*. To put into context of the above-mentioned steps, *SERPINA5* was first prioritized in the differential expression analysis of the

representative phenotype (Step 2a: $p = 0.0001$; Step 2b: false discovery rate [FDR] = 0.0062, log2FC = 1.5). The translational neuropathology approach (Step 4) further identified *SERPINA5* gene expression levels to be monotonically directed ($p = 0.004$, Fig. 5a) and associate robustly with neuropathologic measures (robust association: Braak tangle stage $p < 0.0001$ and FDR < 0.10; Ab39 burden $p < 0.002$ and FDR < 0.22, and Thal amyloid phase $p < 0.001$ and FDR < 0.14, Supplementary Fig. 3). Random forest analyses (Step 5, Fig. 4a, b, and Supplementary Fig. 8) identified *SERPINA5* as the second top predictor in the representative phenotype (mean minimum depth = 2.32, $p < 0.0001$), third top predictor in the extreme phenotype (mean minimum depth = 2.97, $p < 0.0001$), and as the top predictor in the overall discrimination of AD from control (mean minimum depth = 2.18, $p < 0.0001$). Examination of three validation datasets from AMP-AD[25,38,39] (Supplementary Results 2) revealed *SERPINA5* to be significantly upregulated in AD compared to controls in the temporal cortex (Mayo-TCX: FDR = 0.00004, log2FC = 2.1), superior temporal gyrus (Mount Sinai brain bank [MSBB]-Brodmann area [BM]22: FDR = 0.095, log2FC = 1.0), and parahippocampal gyrus (MSBB-BM36: FDR = 0.0016, log2FC = 2.0).

*SERPINA5*, also commonly known as protein C inhibitor, is a glycoprotein that inhibits serine proteases. *SERPINA5* interacts with activated protein C, thrombin, and factor Xa to play a major role in hemostasis/blood coagulation and thrombosis[40–42]. Our studies revealed that in addition to its conventional role in blood coagulation, it may also play a functional role in AD pathogenesis. In both our RNA-Seq and NanoString datasets of hippocampal gene expression, we saw an increase in *SERPINA5* levels among all AD subtypes (Fig. 5a, b). We did not observe differences in *SERPINA5* gene expression levels when males and females were stratified within controls ($p = 0.92$), hippocampal sparing AD ($p = 0.61$), typical AD ($p = 0.83$), or limbic predominant AD ($p = 0.87$) (Supplementary Fig. 12). Linear regression analysis revealed *SERPINA5* expression associated with the advanced marker of tangle maturity ($p = 0.010$) and approached significance with microgliosis ($p = 0.075$) (Figs. 4h, 5c, and Supplementary Fig. 11). To assess SERPINA5 localization within the brain, we utilized immunofluorescence microscopy to examine colocalization. We examined SERPINA5 immunostaining patterns across several brain cell types including astrocytes (GFAP), neurons (MAP2), microglia (resting [IBA1] and activated [CD68]), endothelial cells (CD34), and oligodendrocytes (OLIG2) (Supplementary Fig. 13). SERPINA5 was observed in MAP2-positive neurons in a typical AD hippocampus, whereas SERPINA5 was absent in control brain that lacked tau pathology (Supplementary Fig. 13b). Interestingly, gene expression levels of

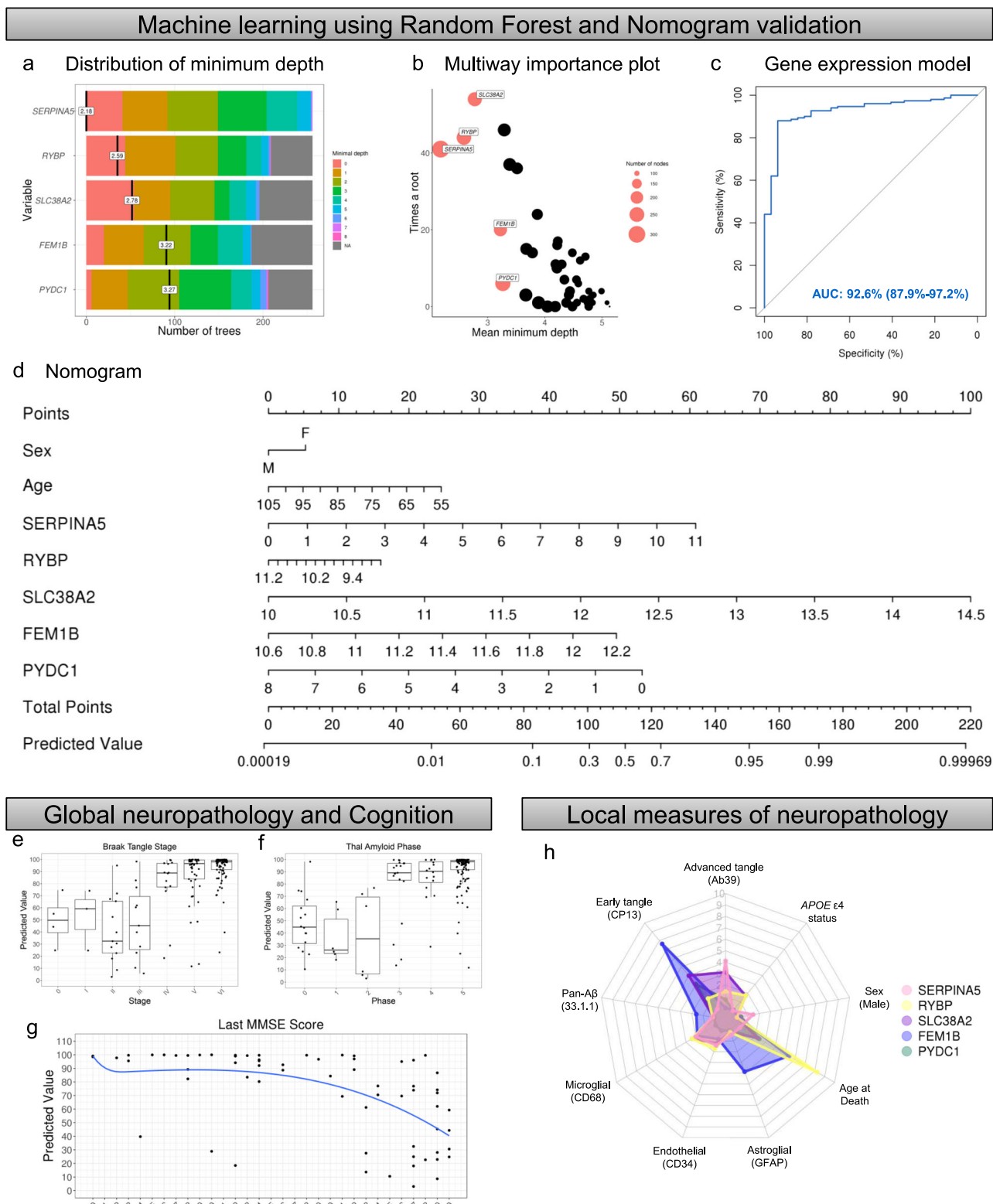

SERPINA5 and the neuronal marker *ENO2* inversely associate in both controls ($R = -0.55$, $p = 0.001$) and AD cases ($R = -0.35$, $p < 0.001$) (Supplementary Fig. 14).

Given SERPINA5's association with neurofibrillary tangle pathology (Fig. 4h) and its predominantly neuronal expression (Supplementary Fig. 13b), we hypothesized that SERPINA5 would follow the corticolimbic patterns inherent to each of the AD subtypes. In an independent cohort (Supplementary Fig. 15), we

employed digital pathology to quantify SERPINA5 immunohisto-chemical burden in hippocampal subsectors (CA1 and subiculum) and association cortices (temporal, parietal, and frontal regions) vulnerable to AD neuropathologic change[7,13] (Fig. 5d). Controls lacked or had minimal SERPINA5 immunopositive lesions (all medians <0.020% burden) and differed in all regions from AD cases ($p \leq 0.002$). As predicted by hippocampal gene expression studies, we observed increasing immunohistochemical burden of

**Fig. 4 Implementation of machine learning to prioritize disease-relevant genes that associate with AD pathology and cognitive decline. a** Minimum depth plot generated using random forest algorithm[34] identified the top five genes predictive of neuropathologically diagnosed AD based on summary of gene position within AD versus control generated trees: *SERPINA5, RYBP, SLC38A2, FEM1B, PYDC1*. **b** Multiway importance plot depicting all genes used in random forest analysis. The top five are highlighted in coral, demonstrating the importance of low minimum depth (x-axis) and high root frequency (y-axis). **c** Gene expression values were built into a logistic regression model along with age and sex to estimate the ability of the top five genes to discriminate between neuropathologically diagnosed AD and controls. Receiver operating characteristic reveals high discrimination with an area under the curve of 92.6%. **d** Nomogram showing the predictive value of neuropathologically diagnosed AD for each of the top five genes along with age and sex. The final predictive value is determined by adding up points from each variable. See Supplementary Fig. 10 for example of utilization. **e** Predictive value of each case and control compared to Braak tangle stage demonstrates higher predictive values corresponding to higher stages. **f** Predictive value of each case and control compared to Thal amyloid phase demonstrates higher predictive values corresponding to higher phases. **g** Predictive value of each case and control with corresponding last MMSE score shows higher cognitive function corresponds with lower predictive value from nomogram. **h** Radar plot depicting the overlay of each of the five genes regressed on demographics and digital pathology measures of tau, Aβ, and cell-specific markers. Radar plot axes correspond to coefficient of partial determination for each of these regressors in the individual models. Additional statistics for radar plot can be found in Supplementary Fig. 11. AD, Alzheimer's disease; AUC, area under the curve; MMSE, Mini Mental Status Examination. Note: Box plots in **e** and **f** are derived from $n = 181$ and $n = 180$ independent samples, respectively. Box plots are displayed at the 25th and 75th percentiles with the median line. Whiskers are drawn to the largest and smallest values that are within 1.5 times the interquartile range from the upper or lower quartile. For observations larger or smaller than this distance, they are shown as observations outside the whiskers.

SERPINA5 among AD subtypes in the hippocampus (AD-specific $p = 0.011$). Based upon our observations of tangle patterns in the cortex of AD subtypes, we hypothesized we would observe an inverse monotonic directionality in the cortex. Thus, we extended our findings to investigate immunohistochemical burden of SERPINA5 across superior temporal, inferior parietal, and mid-frontal cortices (Fig. 5d). The average cortical burden of SERPINA5 demonstrated an overall decreasing monotonic directionality from hippocampal sparing AD > typical AD > limbic predominant AD (AD-specific $p < 0.001$).

Next, we sought to investigate SERPINA5's relationship with neurofibrillary tangle maturity, as more advanced tangle pathology closely associates with neuronal death[9,43]. Neurofibrillary tangles are not static entities[29,44]. They develop progressively from early pretangles with punctate tau staining and perinuclear accumulation to advanced mature tangles with fibrillar structure that form into ghost tangles with diffuse fibrils lacking a nucleus (Fig. 6a)[29]. To confirm the association between advanced tangle pathology (Ab39 burden) and *SERPINA5* gene expression from regression modeling, we performed immunofluorescence microscopy. In pretangles, SERPINA5, like tau, was punctate in nature and found localized to the cytoplasm and perinuclear space (Fig. 6b). A marked increase in SERPINA5 immunostaining was observed in mature tangles aligning with more fibrillar aspects of tau (Fig. 6c). Of note, areas relatively devoid of tau pathology in mature tangles appeared to be occupied by SERPINA5 accumulation. SERPINA5 immunopositive ghost tangles ranged from robustly to lightly stained in their burnt-out form as remnants of tau fibrils (Fig. 6d). SERPINA5 staining was predominantly found in the neuronal soma with only sparse labeling of extrasomal neuritic pathology. SERPINA5 immunopositive neurites could be found proximal to an affected neuron, which represented a subset of tau-positive neurites (Fig. 6b–d). Based upon our findings from immunofluorescence microscopy, we hypothesized that SERPINA5-immunopositive tangles would be in greater abundance in mature tangles and ghost tangles. The number and type of tangle were manually counted in the CA1 subsector of the hippocampus in 60 AD cases on digitized brightfield images of SERPINA5 immunostaining. The proportion of SERPINA5-positive tangles dramatically increased in frequency from pretangles (4%) to mature tangles (52%) and ghost tangles (42%) ($p < 0.001$) (Fig. 6e). In addition to immunopositive tangles, SERPINA5 pathology was found interspersed between and colocalized with tau-positive and thioflavin-S-positive neurites in neuritic plaques (Fig. 6f, g).

Immunofluorescent colocalization of SERPINA5-positive/tau-positive neurofibrillary tangle pathology suggested that SERPINA5 may interact with tau. To investigate this further, we performed a co-immunoprecipitation (co-IP) experiment to determine if we detected a SERPINA5-tau protein–protein interaction. Bulk hippocampal tissue was taken from an AD case and control and used to immunoprecipitate SERPINA5, which was followed by immunoblotting using the tau marker, E1 (ref. [45]). We observed a strong tau signal in the total homogenate (input) of both AD cases and controls, but only detected a tau band in the SERPINA5 co-IP of the AD case (Fig. 6h). This initial study provided supportive evidence that SERPINA5 acts as a tau protein interactor. To provide stronger evidence to support our hypothesis of a SERPINA5–tau interaction, we expanded our studies to the frontal cortex where larger tissue volumes can be dissected for greater yields. Western blot analysis of co-IP experiment revealed that tau was successfully pulled down by SERPINA5 in all three AD cases, but not in controls, suggesting that the SERPINA5–tau interaction is disease-specific (Fig. 6h).

## Discussion

In this study, we leveraged the AD spectrum using both phenotypic differences (extreme phenotype versus representative phenotype)[10–13] and quantitative measures of severity (digital pathology)[10,11,26–28] to identify key proteins involved in the molecular pathogenesis underlying selective hippocampal vulnerability. To our knowledge, this study represents the largest RNA-Seq experiment performed using bulk hippocampal tissue to date[46–48]. Using an approach that combined next-generation transcriptome sequencing with quantitative digital pathology and machine learning, we performed several levels of disease-relevant validation in humans that included: (1) RNA-Seq cohort expansion using NanoString technology[33], (2) machine learning[34] to predict utility of gene expression levels working in concert to discriminate neuropathologic classification, (3) evaluation of the relationship between gene expression levels with macroscopic scales of neuropathology (Braak[7] and Thal[6]) and cognition (MMSE[37]), (4) digital pathology to assess biological relevance of corticolimbic distribution of SERPINA5, and (5) co-IP to examine interaction between tau and SERPINA5. In all, these tools provided invaluable insight into the complexity of hippocampal vulnerability both within the disease spectrum and in AD as a whole, highlighting the role of *SERPINA5, RYBP, SLC38A2, FEM1B,* and *PYDC1*.

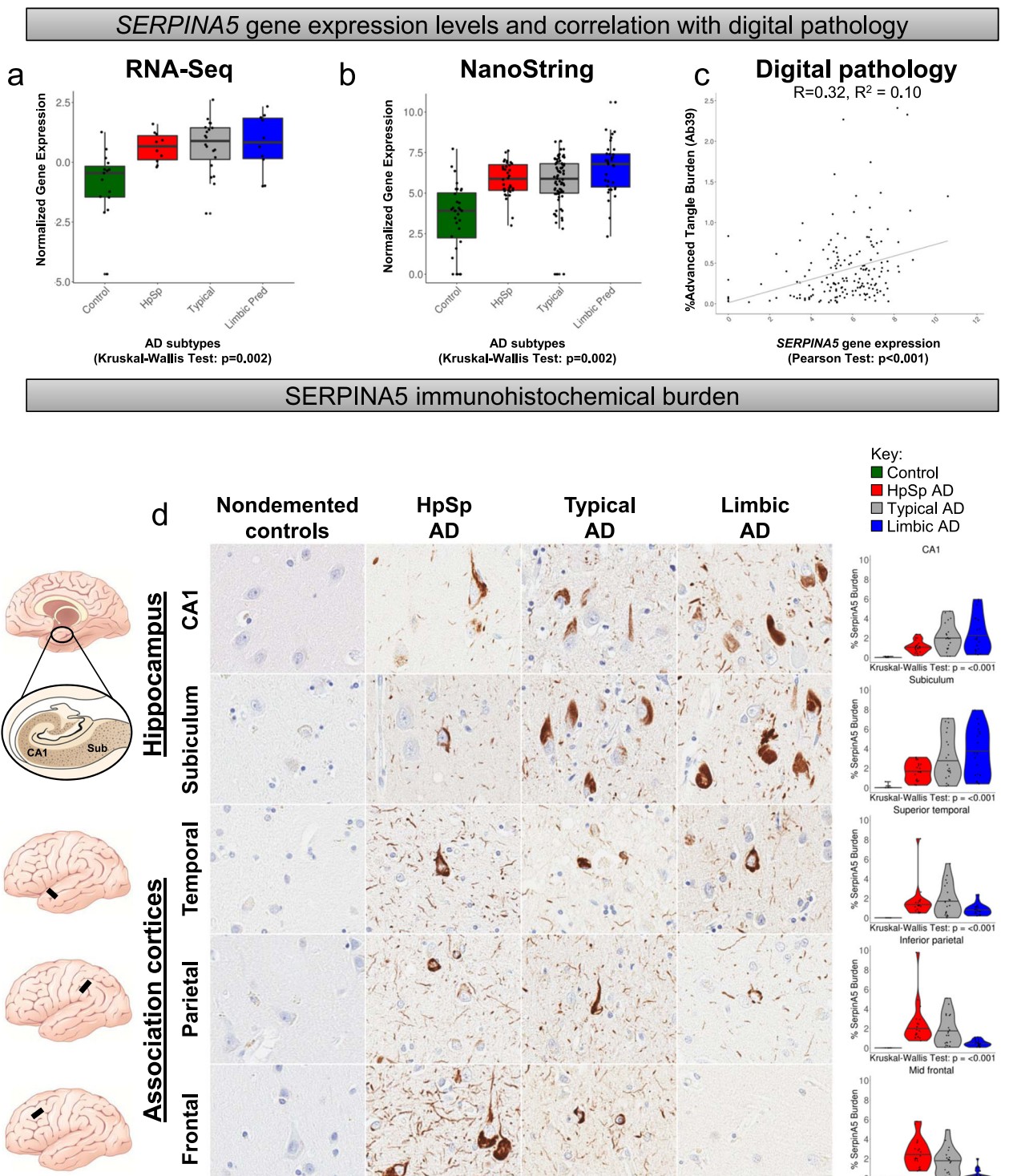

**Fig. 5 SERPINA5 gene expression changes differ among AD subtypes and track with advanced neurofibrillary tangle pathology. a, b** *SERPINA5* expression increases in a monotonic fashion among AD subtypes in the RNA-Seq cohort (**a**), which was replicated in the larger NanoString cohort (**b**). **c** *SERPINA5* gene expression from NanoString cohort associates with % burden of advanced tangle maturity marker Ab39 using two-sided Pearson correlation analysis. **d** Immunohistochemical investigation of SERPINA5 in controls ($n = 10$) and among AD subtypes ($n = 20$ per subtype) reveals a monotonic increase in hippocampal subsectors and a monotonic decrease in the association cortices. Pair-wise comparisons for graphs can be found in Supplementary Fig. 2. Scale bar represents 100 μm. AD, Alzheimer's disease; Aβ, amyloid-β; HpSp, hippocampal sparing; Limbic, limbic predominant; RNA-Seq, RNA sequencing. Note: control cases = green, hippocampal sparing AD = red, typical AD = gray, limbic predominant AD = blue. Box plots in **a** and **b** are derived from $n = 55$ and $n = 182$ independent samples, respectively. Box plots are displayed at the 25th and 75th percentiles with the median line. Whiskers are drawn to the largest and smallest values that are within 1.5 times the interquartile range from the upper or lower quartile. For observations larger or smaller than this distance, they are shown as observations outside the whiskers.

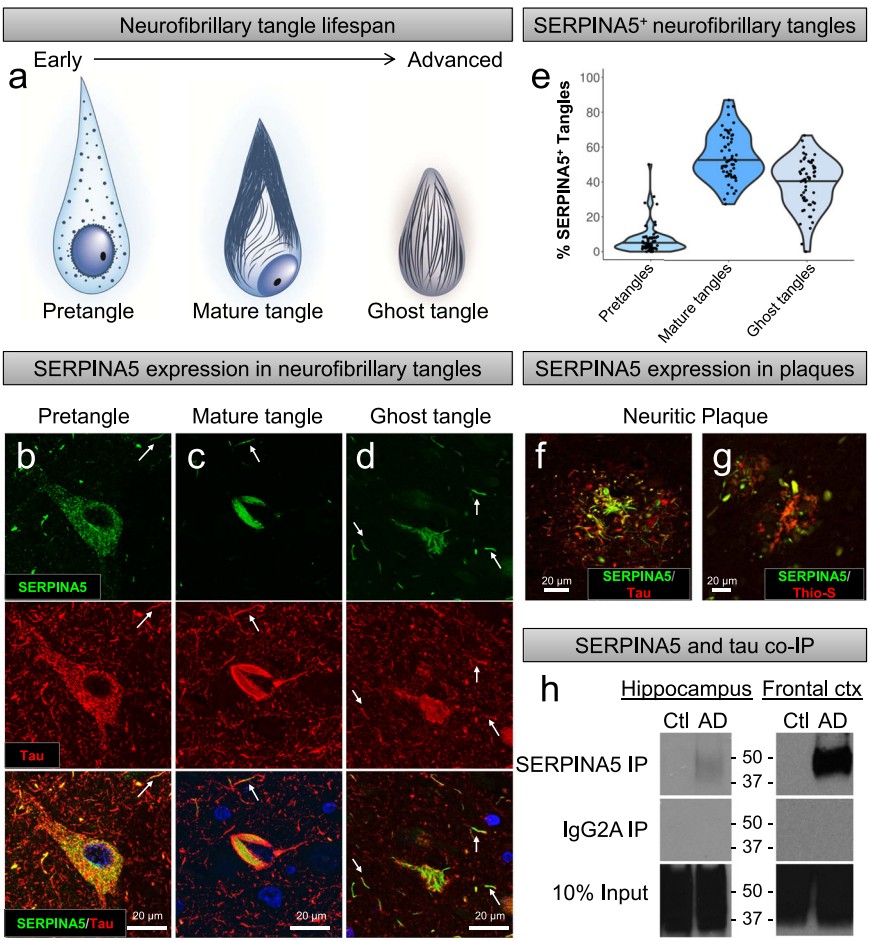

**Fig. 6 SERPINA5 is a tau interactor expressed in neurofibrillary tangles and neuritic plaques. a** Cartoon depiction of the lifespan of neurofibrillary tangle maturity ranging from early to advanced forms: pretangle with punctate tau staining, mature tangle with fibrillar tau, and ghost tangle in which only the remnants of the mature tangle remain once the neuron has died. **b** SERPINA5 accumulates alongside, as well as colocalizes with punctate tau (Tau E178, red) in pretangles. **c** SERPINA5 accumulates in areas fairly devoid of tau and colocalizes along the fibrillar aspect of tau (Tau E178, red) in mature tangles. **d** SERPINA5 accumulates in the extracellular space and to a lesser extent colocalizes with tau (Tau pS396, red) in ghost tangles. SERPINA5 is also observed in a subset of dystrophic neurites (arrows, **b–d**). **e** Manual quantification of SERPINA5-positive neurofibrillary tangle counts in CA1 region of the posterior hippocampus shows a greater frequency of mature tangles and ghost tangles. Pretangles, mature tangles, and ghost tangles were manually quantified. **f, g** SERPINA5 accumulates independently and colocalizes with tau in neuritic plaques as shown by tau (E1) staining and thioflavin-S. Immunofluorescent staining experiments were performed successfully in triplicate for AD and controls. **h** Immunoblot of tau (E1) using immunoprecipitated SERPINA5 demonstrates a tau-SERPINA5 protein complex in the AD brain, but not in control brain. (Left) Tissue was sampled from frozen hippocampi of a 73-year-old male control (Braak = I, Thal = 0) and an 86-year-old male AD case (Braak = V and Thal = 5). (Right) Tissue was sampled from frozen frontal cortices of a 75-year-old female control (Braak = I, Thal = 0) and a 68-year-old male AD case (Braak = VI and Thal = 5). AD, Alzheimer's disease; Ctl, control; co-IP, co-immunoprecipitation; Thio-S, thioflavin-S. Note: uncropped western blots can be found in Supplementary Figs. 19 and 20.

Upon reviewing the published literature, our top five proteins were found to have a multitude of diverse functions within the living system. In addition to SERPINA5's canonical serine protease inhibition function, it exhibits broad protease activity and interacts with phospholipids[49–53], glycosaminoglycans[54], proteins[55–58], and cysteine proteases[59]. These properties make SERPINA5 unique among serine protease inhibitors. RYBP is a multifunctional protein that binds several transcriptional factors[60], mediates histone H2A monoubiquitination[61], and plays roles in development[62,63], apoptosis[64–66], and cancer[60]. SLC38A2 functions as a sodium-dependent amino acid transporter that mediates both the efflux of neutral α-amino acids across the blood–brain barrier and their uptake into neurons[67–69]. FEM1B is an ankyrin repeat protein that induces replication stress signaling through its interaction with checkpoint kinase 1 (ref. [70]), regulates ubiquitin-protein transferase activity[71], and promotes apoptosis[72]. PYDC1 (also known as ASC2/POP1) regulates the innate immune response by suppressing NFκB transcription factor activity and pro-caspase-1 activation[73].

This work underscores the importance of capturing heterogeneity as a phenotype when investigating disease-relevant gene expression changes using bulk RNA-Seq. Our ability to identify the most relevant genes was enhanced by the judicious application of the random forest algorithm[34]. This application of machine learning has wide-ranging utility outside the context of AD, as exampled by its use to uncover transcriptional signatures underlying heterogeneity of smooth muscle cells using single-cell RNA-Seq[74]. To further validate the functional relevance of our findings, we performed regression modeling to assess whether the driver of disease-relevant gene expression changes was underlying neuropathology or cellular admixture. Gene expression levels from our top gene, *SERPINA5*, were upregulated in AD and monotonically directed among AD subtypes. This is in line with a previous study that identified upregulation of *SERPINA5* in the

hippocampus of late onset AD brains[46]. In addition, *SERPINA5* expression was significantly upregulated in AD in all three AMP-AD validation datasets from temporal cortex and para-hippocampal gyrus[25,38,39]. The current study provides further insight demonstrating a strong association between SERPINA5 immunopositive neurons and neuropathologic severity of advanced tangle pathology.

We sought to further investigate cellular distribution of SER-PINA5 based upon a previous study describing SERPINA5 immunoreactivity in microglia[75] and our regression model suggestive of a relationship with microgliosis. Co-immunofluorescence confirmed localization of SERPINA5 to neurofibrillary tangle-bearing neurons in AD brains. In contrast to previous findings in incidental Lewy body disease and Parkinson's disease[75], we did not observe protein expression in non-neural cells. This difference may be due to methodology (immunohistochemical brightfield staining versus co-immunofluorescence) or regional differences (i.e. hip-pocampus versus substantia nigra). Neurons can survive the brunt of early neurofibrillary tangle pathology[76,77], which may represent an invaluable window of time to arrest the neurodegenerative aspects of neurofibrillary tangle progression. Although, SERPINA5-immunopositive structures were only observed in the context of neurofibrillary tangle pathology, gene expression studies revealed an inverse association between *SERPINA5* and the neuronal marker *ENO2* in hippocampi of both controls and AD. This may suggest that upregulation of *SERPINA5* represents a repair process in neurons. Currently, it is unclear if SERPINA5 upregulation is a consequence of failed adaptation to neuronal dysfunction, or a contributor to neurofibrillary tangle formation. Our co-IP experiments provide supportive evidence that SERPINA5 and tau interact in the hippocampus and cortex of AD brains. This was complemented by our microscopy experiments that suggest SER-PINA5 could be a "tipping point" between pretangles and mature tangles. Future studies will be directed at investigating the role of SERPINA5 in neurons with a focus on identifying whether SER-PINA5 acts in a causal or non-causal fashion. This information could aid in identifying a mechanism to prevent or slow neuronal death.

Several limitations in the context of this research should be discussed. At the discovery stage, age and sex were not used to adjust as the AD subtypes exhibit a constellation of demographic and clinicopathologic features. We hypothesize that adjustment of these biological variables may limit identification of important gene expression changes that track with selective hippocampal vulnerability observed in the AD subtypes. As adjustment for age and sex is commonplace in bioinformatic prioritization, future studies lacking distinct clinicopathologic phenotypes should consider adjustment as appropriate to their study. The applied assumption of monotonic directionality was a reflection of the biology of the AD subtypes observed when examining demo-graphic and clinicopathologic differences (e.g., age, sex, *APOE*, cognitive impairment). It is important to note that the utility of this assumption may have excluded genes relevant to hippo-campal vulnerability in AD that did not track with AD subtypes. The monotonic directionality of hippocampal involvement facilitated evaluation of gene expression changes from selective resilience in hippocampal sparing AD to selective vulnerability in limbic predominant AD. Although this range enabled deeper investigation into late-stage AD cases, more work is needed to determine the relevance of these gene expression changes as epiphenomenal versus active players in the pathogenesis of hip-pocampal vulnerability. Furthermore, the selection of literature-based genes does not include several contemporary genes, which may dramatically reduce the value of including known genes that can be leveraged using our approach. To provide a contemporary examination, we applied our translational neuropathology

approach from Step 4 to a recently summarized gene set from Neuner et al.[78] to identify genes that may associate with hippo-campal vulnerability in AD. We identified three additional protein-coding genes that exhibited monotonic directionality among AD subtypes and robustly associated with neuropatho-logic measures: *ADAM10, ANKMY2, ATP5F1* (Supplementary Results 3). We encourage the research community to investigate these and other genes from our NanoString gene list in future studies. Current studies are underway to validate these genes and others prioritized for NanoString in an ethnoracially diverse series to ensure generalizability to all individuals suffering from this devastating disorder.

By embracing heterogeneity of the disease spectrum of AD[10–13,79] and pursuing our hypotheses in the context of the human brain, we have uncovered a tau interactor—SERPINA5. Our study highlights the utility of a multi-disciplinary, team science approach that should be encouraged in all disease-relevant fields. This approach may be especially useful in pro-gressive disorders that accumulate aberrant proteins (e.g. motor neuron disease[80], Parkinson's disease[81]) and spectrum disorders such as diabetes that range in severity[82]. Through the use of sophisticated neuropathologic techniques, we were able to lever-age our understanding of heterogeneity from the perspective of selective vulnerability in AD subtypes[10–13]. Coupled with machine learning, our study highlights the importance of multi-step prioritization efforts to yield promising gene candidates. In this era of big data generation, scientists need advanced methods to prioritize biologically relevant targets from large, hetero-geneous human datasets. Here we outline an approach that offers several aspects of human-relevant validation and provides intri-guing implications into the dynamic aspect of accumulating neurofibrillary tangle pathology that could be targeted to prevent further devastation of the AD brain.

## Methods

**Data reporting**. Sample size was not predetermined using statistical methods.

**Selection of brains from the Mayo Clinic Florida Brain Bank**. A query of the Mayo Clinic Brain Bank identified 1873 neuropathologically diagnosed AD cases with a known AD subtype[13] (Fig. 1). In order to perform gene expression studies, we excluded 534 brains that did not have available frozen tissue for dissection. In an effort to reduce neurobiologic heterogeneity in the remaining 1339 AD cases, we further excluded cases with co-existing pathologies (e.g. Lewy body pathology, tauopathies, tumors). Of the remaining 582 AD cases, we additionally excluded for significant cerebrovascular disease (e.g. infarcts, hippocampal ischemia). Of the remaining 402 AD cases, there were 16 AD cases that self-reported as non-Hispanic white, including two African–American/black decedents and 14 His-panic/Latino decedents. One individual had an unknown ethnoracial status. These 17 AD cases were excluded as each of the AD subtypes was not represented and control brains were not available for matching. Of the remaining 385 AD cases, 68 were hippocampal sparing AD cases, 271 were typical AD cases, and 46 were limbic predominant AD cases. Given the inherent differences in age, sex, and *APOE* ε4 status among the AD subtypes[13], we matched typical cases to each subtype for age and sex prior to tissue dissection. AD cases with the highest RNA quality (RNA integrity number [RIN] ≥ 7) were selected for RNA-Seq with nondemented con-trols selected to represent age distribution and sex. We identified 64 control brains with a Braak tangle stage[7] ≤III that lacked significant neurodegenerative pathology and cerebrovascular disease. Of these, 53 had available frozen tissue for dissection. Upon examination of the frozen brain tissue, 35 had available hippocampus for dissection. Following RNA extraction, three brains were additionally excluded for poor quality RNA. Of the remaining 32 controls, 15 with an RIN ≥7.0 were selected for RNA-Seq analyses.

The final RNA-Seq cohort (*n* = 55) consisted of 10 hippocampal sparing AD, 20 typical AD, 10 limbic predominant AD, and 15 controls (Supplementary Fig. 1). After obtaining RNA-Seq results, these studies were expanded for validation using a custom NanoString codeset of genes on the following: hippocampal sparing AD (*n* = 39), typical AD (*n* = 81), limbic predominant AD (*n* = 40), and nondemented controls (*n* = 32). We note that eight of these cases were excluded from later analysis due to housekeeping gene correction (*n* = 7) or low read counts (*n* = 1). This resulted in a final NanoString cohort containing: hippocampal sparing AD (*n* = 36), typical AD (*n* = 79), limbic predominant AD (*n* = 35), and nondemented controls (*n* = 32) (Supplementary Fig. 4). The final NanoString cohort (*n* = 182)

included the RNA-Seq cohort to enable cross-platform evaluation (Fig. 1). All brains were acquired with appropriate ethical approval from the individual and/or their next-of-kin, and the research performed on postmortem samples was approved by the Mayo Clinic Research Executive Committee (IRB17-007585/Bio00015595).

**Demographics and clinical history**. Age, sex, and *APOE* ε4 status was available from the Mayo Clinic Brain Bank database. Clinical history abstraction was performed retrospectively from existing clinical records sent at the time of brain donation, as previously described[10,83]. The age at onset of cognitive symptoms was calculated by subtracting the date of birth from approximate date at onset and converted to years by dividing by 365.25. Similarly, disease duration was calculated by subtracting the date at onset from date of death. MMSE[37] score (0–30) and test date were abstracted to identify the score closest to death to serve as a measure of perimortem cognition. We report the last MMSE score tested within 3 years of death.

**RNA extraction, purification, and integrity testing**. Total RNA was extracted from dissected hippocampal tissue using TRIzol reagent (Thermo Fisher Scientific 15596026) according to manufacturer's instructions with all steps performed at room temperature (22 °C) unless otherwise stated. About 50–100 mg of frozen tissue was homogenized in 1 mL of TRIzol; 200 μL of chloroform was added to the homogenate, mixed by inverting tubes vigorously for 15 s and left to sit for 3 min. The tubes were centrifuged at $12,000 \times g$ for 15 min at 4 °C. Approximately 400 μL of aqueous layer was transferred to a clean RNase-free labeled tube containing 500 μL of 100% isopropanol, vortexed and incubated for 10 min to precipitate the RNA. Tubes were then centrifuged at $12,000 \times g$ for 10 min at 4 °C. Isopropanol supernatant was discarded and RNA pellet was washed with 1 mL of RNase-free 75% ethanol, vortexed and centrifuged at $7500 \times g$ for 5 min at 4 °C. Ethanol was discarded and RNA pellet was air dried for 5–10 min avoiding complete drying of the pellet. 100 μL of nuclease-free water (Ambion AM9938) was added to each tube and gently vortexed to dissolve RNA.

Total isolated RNA concentration was determined using a NanoDrop 1000 spectrophotometer (Thermo Scientific), and 30 μg maximum of each RNA sample was then treated with DNase I (Qiagen 79254) digested on-column to remove contaminating genomic DNA. Specifically, 100 μL of RNA sample was added to 350 μL of lysis buffer RLT/1% β-mercaptoethanol and mixed well. Then, 250 μL of 100% ethanol was added and then the entire 700 μL sample mix was pipetted onto a RNeasy Mini spin column (Qiagen #74106). The sample columns were centrifuged at $\geq 8000 \times g$ for 15 s at room temperature and the flow through was discarded (all spins are performed at this speed, duration, and temperature); 350 μL of wash buffer RW1 was added to the spin columns and the flow through was discarded, 80 μL of the DNase/RDD working solution was directly added to the spin column membrane and incubated at room temp for 15 min, then briefly centrifuged for 5 s. Again, 350 μL of RW1 was added to wash the columns and the flow through was discarded; 500 μL of RPE buffer was then added to the spin columns and the flow through was discarded. Again, 500 μL of RPE buffer was added to the spin columns and then they were inverted a few times to remove any remaining guanidine thiocyanate from the under the lids. The sample columns were centrifuged at $\geq 8000 \times g$ for 2 min at room temperature and the flow through/collection tubes were discarded (replaced with clean collection tubes and repeat drying for 1 min, then again discarded flow through/collection tubes). The RNA was eluted by placing the spin columns in RNAse-free 1.5 mL tubes and adding 100 μL of RNase-free $H_2O$. The samples were incubated for 2 min then centrifuged at $8000 \times g$ for 1 min at room temperature. The high-purity total RNA was stored at $-80$ °C until use.

The total purified, DNase-treated RNA concentrations were determined using NanoDrop 1000 spectrophotometer (Thermo Scientific) and verified that the 260/230 ratio for samples were at or above 1.8. RNA quality was assessed by calculating RIN and distribution values (DVs) using RNA 6000 Nano Chips on the 2100 bioanalyzer (Agilent). RIN for a sample was determined by several factors, most importantly the calculation of total RNA ratio (proportion of the area under the 18S and 28S rRNA peaks to the total area under the curve) and the height of the 28S peak. RIN values can range from 1 to 10, with 1 representing totally degraded RNA and 10 representing high-quality intact RNA[84]. RIN values for our RNA-Seq cohorts were $\geq 7$. Controls had a median RIN = 7.6 (range 7.2–8.2), hippocampal sparing AD median RIN = 7.3 (range 7.2–8.0), typical AD median RIN = 7.4 (range 7.1–7.9), and limbic predominant AD median RIN = 7.6 (range 7.2–7.8). The NanoString nCounter assay detects very low mRNA concentrations even in significantly degraded RNA samples. As a result, NanoString nCounter de-emphasizes the use of RIN and emphasizes the use of RNA DV, which assesses the proportion of intact RNA fragments with greater than 300 nucleotides (DV300). For our NanoString nCounter studies, we followed best practices and selected cases and controls that had a DV300 >50%[85].

**RNA sequencing**. RNA-Seq analysis was performed using 200 ng of total RNA. Ribosomal RNA depletion and library preparation were performed using TruSeq Stranded Total RNA Library Prep Gold (Illumina 20020599, San Diego, CA). The quality of the library was assessed using Agilent Bioanalyzer before sequencing.

RNA library was sequenced at three samples per lane on the Illumina HiSeq2500 to generate 101 base pairs (bp) × 101 bp paired-end reads.

**Bioinformatic analysis of RNA-Seq data**. MAP-RSeq v2.0 (ref. [86]), an integrative bioinformatics pipeline, was used to obtain gene read counts and various quality-control matrices. More specifically, raw reads were aligned to human reference genome build hg19 using TopHat v2.0.12. Reads mapped to known genes were obtained using featureCounts program in Subread tool kit v1.4.4. The number of reads mapped to known genes was $46 \pm 17$ million in these samples (Supplementary Fig. 16). Read count data were normalized using the R package cqn v1.16.0 (ref. [87]), taking into consideration the library size, gene length, and GC content of each gene coding region. No outliers were detected according to principle component analysis and hierarchical clustering of normalized gene expression data. Source of variation analysis in Partek® Genomics Suite® software[88] v6.6 identified that RIN contributed a significant proportion to gene expression variation on average. Genes with zero read counts in all samples were excluded from downstream analyses. This included 6823 genes out of 57,773 total genes. Genes with a mean normalized expression (in scale of log2 Reads Per Kilobase of transcript, per Million mapped reads) below zero were marked as low expressing and filtered out from further analyses.

**Selection of literature-based genes**. Known genes associated with clinical expression of AD were selected (Fig. 2 [Step 1]). Genes were selected from large consortium efforts. We evaluated 25 genes that are located at loci identified in GWAS of late-onset AD: *ABCA7, APOE, BIN1, CD2AP, CD33, CLU, CR1, EPHA1, MS4A4E, MS4A6A, PICALM, CASS4, CELF1, DSG2, FERMT2, HLA-DRB1, HLA-DRB5, INPP5D, MEF2C, NME8, PTK2B, RIN3, SLC24A4, SORL1, ZCWPW1* (refs. [14–19]). The remaining literature-based genes included those associated with early-onset AD (*APP, PSEN1, PSEN2*)[20,21] or those implicated in modifying the phenotype of AD (*GRN, SIRT1, TOMM40*)[22–24].

A recent review by Neuner and colleagues[78] highlighted 112 genetic loci associated with AD (Supplementary Results 3). Of the 112 genes, one gene (*ACE*) had two available Ensemble IDs and four genes were not detected in our dataset. Of the 108 genes with Ensemble ID's available for post-hoc comparison in our RNA-Seq dataset, three additional genes would have passed our criteria: *ADAM10, ANKMY2*, and *ATP5F1*. The literature gene *PSEN2* is included in the current study.

**Gene differential expression and enrichment analysis for RNA-Seq data**. R edgeR package v3.12.0 (ref. [89]) was used to analyze differentially expressed genes within the representative phenotype (typical AD compared to control) and within the extreme phenotype (limbic predominant AD compared to hippocampal sparing AD) (Fig. 2 [Step 2]). A generalized linear model, adjusting for RIN, was applied with the underlying distribution set as a negative binomial to obtain log2FC and *p*-value of gene expression differences between the above-mentioned phenotypic groups. Given the biological relevance of age, sex, and *APOE* ε4 status to AD subtypes[10–13], these covariates were not used to adjust RNA-Seq data. Briefly, glmFit function from edgeR v3.12.0 (ref. [89]) was applied to fit a negative binomial generalized log-linear model using raw read counts, an offset matrix, dispersion parameters, and design matrix. Next, glmLRT function was used to conduct a likelihood ratio test for differential expression. The offset matrix that incorporated effects of library size, gene length and GC contents was computed using R cqn package v1.16.0 (ref. [87]). The gene wise dispersion parameters were estimated by edgeR functions estimateGLMTrendedDisp and estimateGLMTagwiseDisp. We used an unadjusted *p*-value cutoff of 0.01 to nominate top differentially expressed genes. To enable a broad set of differentially expressed genes to be selected for downstream prioritization efforts utilizing neuropathology measures, bioinformatic prioritization employed a less stringent cutoff (FDR < 0.25) than current convention (i.e., FDR < 0.10). To further enrich for genes found to be downregulated or upregulated by a factor of 2 or higher, log2FC >1 or <−1 cutoff was applied.

Enrichment analysis of differentially expressed genes from Step 2a was performed to identify enriched pathways and gene ontology (GO) terms using GeneGo MetaCore Build 6.24.67895 from Clarivate Analytics (Fig. 2 [Step 3]). Significantly perturbed networks were identified by mapping differentially expressed genes from each phenotype onto pre-built process networks. Process networks were sorted by *p*-value and the top ten statistically enriched process networks were reported.

**Monotonic directionality and neuropathologic association of RNA-Seq data**. We identified genes with a monotonically directed pattern of upregulation or downregulation as follows: control, hippocampal sparing AD, typical AD, limbic predominant AD (Fig. 2 [Step 4]). The directionality was chosen based upon established evidence of hippocampal involvement, where the control was expected to have the least involvement and limbic predominant to have the most severely affected hippocampus[10–13]. An analysis of variance (ANOVA) with RIN as covariate was performed using Partek[88] v6.6 to identify group-wise differences across all four groups. Genes were selected if the overall *p*-value from ANOVA was ≤0.05 and the mean expression monotonically increased or decreased along the spectrum: control, hippocampal sparing AD, typical AD, and limbic predominant AD. Gene

class assignment of protein-coding was used to further prioritize monotonically directed genes.

Using the linear model (lm) function in R, linear regression was performed to examine the relationship between gene expression and neuropathologic measures of tau and Aβ (Fig. 2 [Step 4]). Specifically, gene expression was regressed on tau measures including Braak tangle stage, an early marker of tangle maturity (CP13, directed at phospho-S202), and an advanced marker of tangle maturity (Ab39, conformational epitope). Gene expression was regressed on Aβ measures including Thal amyloid phase[6] and a pan-Aβ marker (33.1.1, raised against Aβ1-16). RIN was included as a covariate for each of the above regression models and run separately for each tau and Aβ measure. Regression analyses for genes identified in Steps 1–3 were examined across all four groups (control, hippocampal sparing AD, typical AD, limbic predominant AD) and within the representative phenotype (control, typical AD) (Supplementary Fig. 3). Genes found to robustly associate in both analyses were further selected for NanoString analyses.

**NanoString nCounter™ analysis.** To investigate the biological significance of the prioritized genes in (Fig. 2 [Step 5]), 150 ng of purified total RNA was used for NanoString nCounter v1.0.84 gene expression analysis. A custom codeset of 56 genes was designed to validate RNA-Seq findings in the expanded NanoString cohort of hippocampal tissue from 190 brains. The 56 genes included the 45 genes prioritized in Step 4 and 12 housekeeping genes. The 12 housekeeping genes were selected as they previously demonstrated stable expression across a diverse series of human tissue[90]. These included *C1orf43, CHMP2A, EMC7, GPI, PSMB2, PSMB4, RAB7A, REEP5, SNRPD3, VCP, VPS29*, as well as the traditional reference gene *GAPDH*. As one gene (*CFI*) was inadvertently left off of the custom codeset, the remaining analyses were conducted on 44 of the 45 prioritized RNA-Seq genes.

For NanoString nCounter analysis, R package OSAT[91] v1.18 was used to perform blocking randomization on 192 samples across all four groups (control, hippocampal sparing AD, typical AD, and limbic predominant AD), sex (male or female), age (55–103), and RIN (4.6–8.9). Each NanoString plate had 2 × 6 wells and held 12 samples. Thus, age and RIN were stratified into 15 levels each. Samples were assigned to 15 plates in such a way that 13 plates held 12 samples each and two plates held 11 samples each. Samples were distributed in each plate as evenly as possible according to an objective function. The tests of independence between plates and sample variables (diagnosis, sex, age, and RIN) were not significant. Manual check also revealed that the number of subjects in any specific diagnosis group differed at most by 1 across plates, and mean values of age and RIN were similar across plates. To test technical reproducibility, we plated replicate RNA extracted from the brains of a hippocampal sparing AD and from a limbic predominant AD. Thus, 190 brains were assessed, but 192 samples were plated. Duplicate concordance was 98.4% for the hippocampal sparing AD and 98.1% for the limbic predominant AD.

NanoString gene expressions were assayed using the NanoStringNorm[92] package v1.2.1 in R v3.4.2. Samples were normalized twice, first for quality control and second for background correction using (Eq. 1):

$$c \times \left(\frac{m}{s}\right)$$

where *c* was the count data, *m* was the geometric mean of the sum of the housekeeping genes across samples, and *s* was the sum of the housekeeping genes for a given sample. Samples with content greater than three standard deviations were removed during the first pass (quality control) and samples requiring a 2.5-fold adjustment after visual assessment were removed during the second pass (background correction). Seven cases were excluded from further analyses as they were identified to have a greater than 2.5-fold adjustment following housekeeping normalization (Supplementary Fig. 5). One case was excluded due to low gene expression counts overall. The final NanoString cohort consisted of 182 cases and controls.

**Random forest modeling.** After selection of the 44 candidate genes from Steps 1-4, additional prioritization of the candidate genes (Fig. 2 [Step 5]) was required to facilitate further basic discovery research. Given the possibility of complex interactions among the genes we used a random forest model (Supplementary Fig. 7), which is a non-parametric, rules-based approach[34]. The set of classification trees was developed using the randomForest package v4.6-12 (ref. [93]) in R v3.4.2. The default tuning parameters were accepted as the goal was to provide a preliminary investigation into how effectively the gene expression values could discriminate typical AD from controls in the representative phenotype, and discriminate limbic predominant AD from hippocampal sparing AD in the extreme phenotype. AD in the context of the final modeling was any of the three subtypes to enable a spectrum of hippocampal vulnerability to be captured. Thus, the final top five genes were derived from random forest analyses run to discriminate all AD cases from controls. Five top genes rather than six or ten were arbitrarily chosen based on prioritization and feasibility for downstream applications, not based on statistical selection. Variable importance was assessed using the R package randomForestExplainer v0.9 (ref. [94]). This package enabled exploration of variable inclusion metrics, assessment of changes in model performance with exclusion of a variable, and interaction summaries within the data. Two variable inclusion metrics were considered of direct interest.

First, the number of times a variable was selected as a root was considered (node = 0, Supplementary Fig. 7). This metric tabulates the number of times a variable was selected as a root across the 500 trees. A higher number suggests that among the randomly selected subsets of genes chosen to create each of the 500 trees, the gene consistently provided the highest degree of precision. The second metric, minimum depth, builds upon the number of times a gene was selected as a root variable by considering where, in terms of depth of the tree, a gene was located. The closer they were to the root, the more important they were at the classification task. The minimum depth tabulates all the different positions each variable takes across all the trees in the forest. A smaller number indicates that the variable was chosen closer to the root if it was not the root. These two metrics can be plotted as a scatter plot (multiway importance plot) to understand how genes may interact. While many of the genes were found to be associated with AD status using the random forest approach (Supplementary Results 1), the top five genes with smallest minimum depth were selected to examine biological relevance.

**Random forest model discrimination and nomogram.** In the previously described random forest model, only gene expression values were used as candidate AD classification variables. To better understand how discriminatory the prioritized genes were in classifying AD from control, a series of multiple logistic regression models were considered. The base model was the gene expression value only. A second set of models provided adjustment for age and sex. Odds ratios (OR) and area under the receiver operating curve (AUC) were used to quantify the degree to which the variables discriminated AD from control. To present the model results succinctly in a visual form (Fig. 4d), a nomogram was developed[35,36]. This visual model allows one to assess relative importance of variables from the scale of the points associated with observed values and to quickly perform the non-linear calculation of probabilities using simple addition. To estimate the probability of an AD case having a higher predicted value from the nomogram compared to a control case, we evaluated a receiver operating characteristic (ROC) curve (Fig. 4c). We applied a conservative cutoff of 77.025, given the greater proportion of AD cases (*n* = 150) compared to controls (*n* = 32). This cutoff value provides the "line" at which the model performs the best in terms of sensitivity, specificity, and accuracy when classifying AD versus Control. At 77.025, our sensitivity was 0.88 (95% CI: 0.83–0.93), our specificity was 0.94 (95% CI: 0.85–1), and our accuracy was 0.89 (95% CI: 0.889–0.891).

**AMP-AD RNA-Seq validation datasets.** We downloaded gene read counts and metadata files of three RNA-Seq cohorts from AD knowledge portal on Synapse (www.synapse.org, Supplementary Fig. 17a), each consisting of postmortem brains from AD and controls (Supplementary Fig. 17b). These included the Mayo Clinic temporal cortex (Mayo-TCX)[25,38], and Mount Sinai VA Medical Center Brain Bank superior temporal gyrus (MSBB-BM22), and parahippocampal gyrus (MSBB-BM36)[39]. The gene read counts were generated through a consensus bioinformatics pipeline that aligned raw RNA-Seq data files, counted reads mapped to genes, and reported quality control measures. Further details can be found on the AD knowledge portal (https://adknowledgeportal.synapse.org, Synapse ID: syn17010685) and by Wan et al.[95]. Neuropathologic information provided in the available metadata files was used to assign individuals as AD, control, or other (Supplementary Fig. 17b). AD cases and controls were the focus of our differential expression analyses. As previously described in the "Gene differential expression and enrichment analysis for RNA-Seq data" section, we performed differential expression analysis between AD and control samples using R edgeR package v3.12.0 (ref. [89]) and cqn package v1.16.0 (ref. [87]). The covariates included in the design matrix were diagnosis (AD or control, categorical), RIN (continuous), age at death (continuous), sex (categorical), and flowcell (categorical) for all four datasets (including current study), plus the source of samples (categorical) for Mayo-TCX dataset. The resulting sample sizes for Mayo-TCX was *n* = 68 controls and *n* = 80 AD, MSBB-BM22 was *n* = 33 controls and *n* = 70 AD, MSBB-BM36 was *n* = 30 controls and *n* = 56 AD, and the current study was *n* = 15 controls and *n* = 20 typical AD (Supplementary Fig. 17c and Supplementary Results 2).

**Immunohistochemistry.** Routine sampling of the posterior hippocampus was performed at the level of the lateral geniculate at the time of brain cutting[96,97]. Formalin-fixed tissue was paraffin-embedded for archival purposes. Serial sections of the posterior hippocampal tissue blocks were cut at 5-μm and mounted on glass slides. Slides were deparaffinized using three 5-min xylene washes followed by two 3-min 100% ethanol washes and one 3-min 95% ethanol wash. Immunohistochemical staining of tau, Aβ, and cellular markers was performed on a Dako Autostainer (Universal Staining System, Carpinteria, CA) using 3,3′-diaminobenzidine (DAB) as chromogen. Antibodies to an early marker of tangle maturity (CP13, 1:1000)[31], an advanced marker of tangle maturity (Ab39, 1:350)[98], pan-Aβ (33.1.1, 1:1000), astrocytic marker (GFAP, 1:5000), endothelial marker (CD34, 1:25), and activated microglia (CD68, 1:1000) were used (Supplementary Fig. 18a).

**Digital pathology.** Digital pathology enables scanning, naming, tracing, and analysis of serially sectioned immunohistochemically stained tissue[10,11,13,26]. Microscope slides stained with the antibodies outlined in Supplementary Fig. 18a were digitally scanned using an Aperio AT2 system (Leica Biosystems, Buffalo

Grove, IL) and annotated using Aperio ImageScope v12.4.2.7000 (Leica Biosystems, Buffalo Grove, IL). The hippocampus proper was traced to enable direct comparison with gene expression data from the frozen hippocampus dissected from the contralateral side. Neuroanatomical landmarks were used during tracing to establish the temporal horn as the lateral border and the white matter of parahippocampal gyrus as the inferior border. The fimbria was not included in tracing of the hippocampus. Slides were analyzed with custom-designed macros using the color deconvolution algorithm (CP13, CD68) or positive pixel count algorithm (Ab39, GFAP, CD34, 33.1.1). Data obtained from the color deconvolution and positive pixel count analysis were a percent burden that represented the percent area of staining out of the area annotated. Digital pathology measures were performed in those with sufficient tissue for serial immunohistochemical staining ($n = 54/55$ for RNA-Seq cohort and $n = 170/182$ for NanoString cohort). Supplementary Fig. 18b contains custom input information for macro development.

**SERPINA5 immunohistochemical validation.** Following identification of *SERPINA5* as our top gene, biological relevance was investigated immunohistochemically in an independent series of 10 controls, 20 hippocampal sparing AD, 20 typical AD, and 20 limbic predominant AD cases (Supplementary Fig. 15). The posterior hippocampus, superior temporal cortex, inferior parietal cortex, and mid-frontal cortex were cut in 5-µm sections and immunostained with a SERPINA5 antibody (1:100, mouse IgG, R&D Systems, Minneapolis, MN) using the Lab Vision Autostainer 480S (Thermo Scientific, Waltham, MA) and DAB (Dako, Carpinteria, CA) as chromogen. Immunostained sections were similarly digitized with Aperio technologies. The pyramidal layer of the CA1 and subiculum within the hippocampus were traced separately for analysis. The lacunosum layer marked the superior boundary and the alveus indicated the inferior boundary for tracing. Additionally, the midway of the dentate gyrus was used to operationalize the CA1-subiculum border. The cortical regions were traced in the area with the most pathology along the strait of the gyrus. To analyze the traced regions, a color deconvolution algorithm was custom-designed based upon the tinctorial properties of immunopositive lesions and set conservatively to avoid quantifying lipofuscin pigment. Data obtained were a percent burden of SERPINA5 per total area annotated.

**SERPINA5 tangle counts.** Using Aperio's Image Scope v12.4.3.7001, the CA1 region of the hippocampus was outlined as stated in the digital pathology methods section. The counter tool was used to mark tangle-bearing neurons in the 20 hippocampal sparing AD, 20 typical AD, and 20 limbic predominant AD cases used above. Tangle maturity levels were identified based on the following set of rules: (1) pretangles exhibit diffuse/granular staining, stain less intensely than mature tangles, and contain a nucleus; (2) mature tangles exhibit intense immunostaining (generally throughout the entire neuron) and contain a nucleus; (3) ghost tangles exhibit less intense immunostaining than mature tangles, loosely arranged bundles of fibers, and no nucleus.

As tangles occur over a lifespan of maturity[29], we also categorized two intermediary stages: (1) "Intermediary 1" consisted of the level between pretangles and mature tangles. These tangles have intense diffuse or focal granular immunostaining, as well as some fibrillary staining pattern and contained a nucleus. (2) "Intermediary 2" was the level between mature tangles and ghost tangles and was identified by intense immunostaining with no nucleus. If tangles did not fit nicely into the above categories based on shape, immunostaining, or nuclear position, these tangles were considered "unclassified" and were not included in analysis. Furthermore, tangles were not counted if the nucleus lay outside the annotated region or if less than half of the body of the tangle lay inside the annotated region. Tangles were reviewed with neuropathologists (D.W.D., M.E. M.) and edited for all samples.

For analysis, Intermediary 1 counts were combined with mature tangles counts since they did not meet the requirements for pretangles. Similarly, Intermediary 2 counts were added to ghost tangles counts since they did not meet the requirements for mature tangles. Total tangles represent the number of pretangles, mature tangles, and ghost tangles in the annotated CA1 region. Percent pretangles, mature tangles, and ghost tangles were then calculated by multiplying by 100.

**Immunofluorescent staining.** Formalin-fixed, paraffin-embedded posterior hippocampus sections were deparaffinized as described in the previous section. Antibody retrieval was performed by steaming slides for 30 min in either deionized water or citrate buffer (Dako S1699). The tissue was blocked in Dako protein block (Dako X0909) for 1 h at room temperature. Primary antibody was diluted in Dako diluent (Dako S3022) at appropriate concentration (see Supplementary Fig. 18c) and allowed to incubate at 4 °C overnight. The slides were washed three times with PBS + 0.1% Tween-20 then incubated in appropriate secondary plus DAPI (1:500, Thermo D1306) for 4 h at room temperature. After secondary incubation (see Supplementary Fig. 18d), slides were washed three times with PBS + 0.1% Tween-20 and autofluorescence was quenched using 1% Sudan Black B (Sigma 199664) for 5 min. Slides were coverslipped with AquaPolyMount (Polysciences 18606) and imaged using a LSM 880 confocal microscope with AiryScan (Carl Zeiss Microscopy). Immunofluorescent staining experiments were performed successfully in triplicate for AD and controls.

**Co-IP and western blot.** Experiments were performed using Dynabeads Co-IP kit (Invitrogen 1432D). Posterior hippocampus from frozen brains was dissected and weighed. The tissue was then homogenized in IP buffer (IP buffer plus 100 mM NaCl, 25 mM MgCl$_2$, 1 mM DTT, 1:200 PMSF) and allowed to incubate for 15 min. After spinning down at $2600 \times g$ for 5 min, samples were brought to a final concentration of 0.11 g/mL. Beads were conjugated to SERPINA5 (R&D System MAB1266) antibody. Antibody-conjugated beads were then added to the homogenate and allowed to incubate for 1 h at 4 °C. Beads were washed and sample was eluted as per manufacturer's instructions.

Samples were mixed in a 3:1 ratio with Laemmli buffer (Bio-Rad 161-0747) with 10% BME (Sigma M6250) and boiled at 90 °C for 10 min. Samples were loaded onto a 10% Mini-PROTEAN®TGX™ Precast Protein Gels (Biorad 4561034) as well as 10 µL of precision plus dual color standards ladder (Biorad 1610374). Gel was run at 100 V for 2 h on ice. Transfer was performed using Trans-Blot® Turbo™ RTA Mini Nitrocellulose Transfer Kit (Biorad 1704158). The membrane was blocked using 5% dry milk in TBST for 1 h before incubating in primary antibody diluted in 5% dry milk in TBST overnight at 4 °C. The membrane was washed several times using TBST then incubated in secondary antibody for 1 h at room temperature. Secondary antibodies were as follows: Peroxidase AffiniPure F(ab')$_2$ Fragment Donkey Anti-Rabbit IgG (H+L) (Jackson Immunoresearch 711-036-152) or Peroxidase AffiniPure F(ab')$_2$ Fragment Donkey Anti-Mouse IgG (H+L) (Jackson Immunoresearch 715-036-150). Membranes were treated with SuperSignal West Pico Chemiluminescent substrate (Thermo 34080) and imaged using Kodak X-OMAT 2000 Processor with BioMax light film (Carestream 178-8207).

Tissue was sampled from frozen hippocampi of a 73-year-old male control (Braak = I, Thal = 0) and an 86-year-old male AD case (Braak = V, Thal = 5). A second set of independent cases and controls was additionally investigated. For these, the tissue was sampled from frozen mid-frontal cortices from three AD cases (#1, 64-year-old female [Braak = VI, Thal = 5]; #2, 60-year-old female [Braak = VI, Thal=4]; #3, 68-year-old female [Braak VI; Thal 4]) and three controls (#4, 75-year-old female [Braak = I, Thal = 0]; #5, 78-year-old male [Braak = II, Thal = 1]; #6, 96-year-old female [Braak = II, Thal = 3]). Raw western blot images can be found in Supplementary Figs. 19 and 20.

**Statistical considerations.** To examine the association of neuropathologic markers with gene expression values, multivariable linear regression models were used to regress the observed gene expression values on digital pathology measures of AD pathology (tau [CP13, Ab39], Aβ [33.1.1]) and cellular diversity (activated microglia [CD68], endothelia [CD34], and astroglia [GFAP]), age at death, male sex, and presence of *APOE* ε4 risk allele. Models were fit, and the coefficients of partial determination were used to quantify the conditional association of each disease marker with the gene expression values. These coefficients of partial determination were combined across models for each gene in the form of a radar plot (Fig. 4h) to allow for visual comparison of how each gene may be differentially associated with neuropathologic markers.

All tests were two-sided and all p-values <0.05 were considered statistically significant. No correction for multiple testing has been applied to p-values to facilitate the application of other thresholds for statistical significance[99]. Continuous variables were summarized with median and range while categorical variables were summarized with frequency and percent. Kruskal–Wallis Rank Sum and Wilcoxon Rank Sum tests with unadjusted post-hoc comparisons of gene expression values across disease phenotypes were used to initially explore the data. All statistical analysis was performed in R Statistical Software v3.4.2 (R Foundation for Statistical Computing, Vienna, Austria).

**Reporting summary.** Further information on research design is available in the Nature Research Reporting Summary linked to this article.

## Data availability

All requests for raw and analyzed data and related materials, excluding programming code, will be reviewed by Mayo Clinic's Legal Department and Mayo Clinic Ventures to verify whether each request is subject to any intellectual property or confidentiality obligations. Requests for patient-related data not included in the paper will not be considered. Any data and materials that can be shared will be released via a Data Use/ Share Agreement or Material Transfer Agreement. The Accelerating Medicines Partnership (AMP-AD) data in this manuscript are available via the AD Knowledge Portal (https://adknowledgeportal.synapse.org). Ensemble database queries used to examine gene names from RNA-Seq files relied upon http://dec2015.archive.ensembl.org/index.html, and for contemporary literature-based gene set derived from Neuner et al.[78], http://aug2020.archive.ensembl.org/index.html was used.

## Code availability

Programming code related to the data preprocessing will be made available under the GNU General Public License version 3 upon request to Z.I.A. (attia.itzhak@mayo.edu). Key scripts for cqn normalization and edgeR are available at https://github.com/gitSpacexw/RNASeq_SERPINA5.

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

## Acknowledgements

This study was supported by the National Institute on Aging (R01 AG054449, U01 AG057195, P01 AG003949, P30 AG062677, P50 AG047266, U01 AG006786, and R01 AG034676), the Florida Department of Health, Ed and Ethel Moore Alzheimer's Disease Research Program (6AZ01, 8AZ06, 20A22), and the Alzheimer's Association (AARG-17-533458). We would like to thank Peter Davies for providing the CP13 antibody, Shu-Hui Yen for Ab39 antibody, Leonard Petrucelli and Casey Cook for the E1 antibody, and Pritam Das for the 33.1.1 antibody. We are grateful to Ariston L. Librero, Jo Landino, and Virginia Phillips for histological support; to Monica Castanedes-Casey for immunohistochemical support; and to our brain bank study coordinators Rachel LaPaille-Harwood and Jessica F. Tranovich. This work would not be possible without the generosity of the Gerstner Family Career Development Award, Center of Individualized Medicine at Mayo Clinic, and a kind gift from David and Frances Strawn. We thank the patients and their families for their generous brain donations to help further our knowledge of Alzheimer's disease. This work was supported by National Institute on Aging (U01 AG046139, RF1 AG051504 to N.E.T.); National Institute of Neurological Disorders and Stroke (R01 NS080820 to N.E.T.). Study data were provided by the following sources: The Mayo Clinic Alzheimer's Disease Genetic Studies, led by Dr. Nilüfer Ertekin-Taner and Dr. Steven G. Younkin, Mayo Clinic, Jacksonville, FL, using samples from the Mayo Clinic Study of Aging, the Mayo Clinic Alzheimer's Disease Research Center, and the Mayo Clinic Brain Bank. Data collection was supported through funding by NIA grants P30 AG062677, R01 AG032990, U01 AG046139, R01 AG018023, U01 AG006576, U01 AG006786, R01 AG025711, R01 AG017216, R01 AG003949, NINDS grant R01 NS080820, CurePSP Foundation, and support from Mayo Foundation. Study data include samples collected through the Sun Health Research Institute Brain and Body Donation Program of Sun City, Arizona. The Brain and Body Donation Program is supported by the National Institute of Neurological Disorders and Stroke (U24 NS072026 National Brain and Tissue Resource for Parkinson's Disease and Related Disorders), the National Institute on Aging (P30 AG19610 Arizona Alzheimer's Disease Core Center), the Arizona Department of Health Services (contract 211002, Arizona Alzheimer's Research Center), the Arizona Biomedical Research Commission (contracts 4001, 0011, 05-901, and 1001 to the Arizona Parkinson's Disease Consortium), and the Michael J. Fox Foundation for Parkinson's Research. The

results published here are in whole or in part based on data obtained from the AD Knowledge Portal (https://adknowledgeportal.synapse.org/, Synapse IDs: syn17010685, syn3163039, syn20801188). These data were generated from postmortem brain tissue collected through the Mount Sinai VA Medical Center Brain Bank and were provided by Dr. Eric Schadt from Mount Sinai School of Medicine.

## Author contributions

M.E.M. designed and supervised the entire study. K.M.H., A.M.L., M.D.T., and M.E.M. dissected frozen brains. K.M.H. performed RNA extraction and quality control. D.W.D. performed neuropathologic evaluation and quantitative thioflavin-S counts. R.C.P., R.D., N.R.G., and N.E.T. performed the neurologic evaluation. A.M.L. abstracted the clinical information. X.W., Y.W.A., X.T., and H.L. performed all bioinformatic analysis. D.S. performed NanoString Analysis. E.R.L. and R.E.C. performed all statistical analysis. K.M.H., N.O.A., and B.J.M. evaluated and validated the antibodies. I.F., S.A.L., and A.M.L. performed the digital pathology. C.M.M. performed the SERPINA5-positive tangle counts. A.M.C. and B.J.M. designed and performed the SERPINA5 experiments and data analysis. M.A., M.M.C., and O.A.R. performed genotyping and provided substantial intellectual contribution. N.E.T. provided funding, direction, and supervision for the generation of Mayo Clinic AMP-AD data and analysis of all AMP-AD data in this manuscript. X.W., Z.S.Q., T.A.P., T.P.C., and M.A. analyzed all AMP-AD data. M.E.M. and A.M.C. wrote the manuscript. All authors edited and reviewed the manuscript.

## Competing interests

R.C.P. is a consultant for Hoffman-La Roche, Merck, Biogen, Eisai. R.C.P. is on the Data safety and monitoring board for Genentech. N.G.R. takes part in multi-center studies funded by Lilly, Biogen, and Abbvie. M.E.M. served as a consultant for AVID Radio-pharmaceuticals. All other authors declare no competing interests.
