## [Peer Review File · Nature Communications]

REVIEWER COMMENTS

Reviewer #1 (Remarks to the Author):

In this study, Crist et al. integrated multiple approaches to identify disease-relevant genes that associated with neuropathologically diagnosed Alzheimer's disease (AD). They specifically show that SERPINA5 is a novel Tau-binding partner in the hippocampus that potentially regulates the maturation of neurofibrillary tangles. The authors first conducted bulk transcriptome profiling in hippocampi from a cohort comprising 40 AD cases and 15 controls to identify AD-associated genes and validated their findings in an expanded NanoString cohort comprising 150 AD cases and 32 controls. The authors integrated bulk RNA sequencing profiling with quantitative Tau and amyloid-beta pathology to uncover genes related to the typical AD phenotype and extreme phenotypes. Furthermore, the authors adopted a machine learning method to identify 5 top candidate genes as neuropathologically diagnosed disease predictors in the human brain. They subsequently showed that one of the top genes, encodes SERPINA5, is a Tau-binding partner.

Overall, this work is straightforward. The topic is significant and of general interest to a broader audience, which is within the scope of Nature Communications. However, in the current study, both the analytical and experimental evidence are not sufficient to substantiate their conclusion. Some crucial analytical and experimental details are missing and the rationale of the analytical approaches is not clear. Specifically, the authors did not provide details on their RNAseq analysis, e.g. whether they consider the age and gender effects of individuals. Furthermore, the selection criteria of the differentially expressed genes associated with AD subtypes are not stringent enough. Specifically, when using edgeR to determine the significantly differentially expressed genes, the authors used a filtering FDR < 0.25. As the commonly used cutoffs of filtering criteria are < 0.1, is there a reference to support this filtering criterion? Also, there is a lack of details on the approaches of identifying the differentially expressed genes associated with AD phenotypes (Fig. 2; Step 1 and 3). In particular, the authors did not include the rationale of how the 31 genes were selected based on literatures (Fig. 2; Step 1), and the details of how the cellular processes and the corresponding genes were identified (Fig. 2; Step 3). Also, in the current study, an independent validation cohort was not used to validate the performance of the prediction model. A separate dataset for validation should be included. While the authors showed that SERPINA5 expression is associated with the AD subtypes and the maturation of neurofibrillary tangles, there is insufficient evidence showing this protein is a Tau-binding partner (the interaction of SERPINA5 and Tau was indeed barely detectable in AD brain Fig. 6H), and whether it is involved in tangle maturation.

Specific points:

Regarding the differential gene expression analysis, despite providing RINs, the authors need to clarify whether they adjusted for covariates, such as age, sex, genetic effects, etc.

When the authors distinguished AD from controls by using the prioritized genes identified in the training dataset, the validation dataset was not independent from the training dataset. A separate data set for validation should be included.

It is important to determine whether the 5 AD genes identified are specific to AD. Thus, the predictive values of these genes for other diseases (at least for some neurodegenerative diseases) should be evaluated.

The rationale and data for the selection of SERPINA5 should be clearly presented.

While the top 5 predictive genes are presented for each comparison in the prediction model (Extended Data Fig. 8), it is unclear how many genes passed the threshold (i.e., "high-predictive genes") and how many "high-predictive genes" overlapped between the typical and extreme phenotypes.

Minor points:

Figure 1b: Need to ensure terminology is used consistently: "RNA-Seq cohort" and "NanoString Cohort."

Figure 3a: Regarding the expression of DE Genes, " $1 \geq \log FC \leq -1$," the labelling is incorrect. It should be $\log FC > 1$ or < -1 .

Reviewer #2 (Remarks to the Author):

The manuscript from Crist and colleagues leverages an extreme phenotype design and comprehensive neuropathological assessment to identify genes that may be associated with tau dysregulation in Alzheimer's disease, focusing particularly on the hippocampus. The study design is creative and makes use of incredibly detailed neuropathological characterization to carefully define each study group. The small sample size is then overcome by leveraging annotation and bioinformatic techniques to improve statistical power, a replication dataset leveraging the emerging NanoString platform, and multi-modal validation to confirm the top candidate including co-IP using microscopy. The results implicate SERPINA5 as a novel tau binding partner worth pursuing in experimental follow-up to clarify causality, while also raising 4 additional novel candidate genes relevant to tau changes in AD. Overall, this is a very strong study and the SERPINA5 results are quite compelling. Some limitations that could strengthen the manuscript are listed below.

Major Concerns

(1) Given the prioritization approach taken, each step of gene selection is critical and only some steps are well described. The selection of AD loci for example includes proximal genes even when the functional gene in the region is known to be elsewhere (the inclusion of CELF1 instead of SPI1 is one example). Meanwhile many additional causal genes that are well validated are not included (ADAM10, TREM2, UNC5C, PLGC2, PILRA). The gene symbol presented in the GWAS papers is rarely a functionally validated gene, but many of the regions now do have functionally validated genes that can be leveraged in this type of analysis. The poor gene list that is included in the first step undermines the downstream processes and dramatically reduces the value of including known AD genes. There is a nice review that recently came out that provides a quality summary of the known v. unknown genes at AD loci from the GWAS papers:
<https://www.sciencedirect.com/science/article/pii/S0969996120302515>.

(1a) A related challenge is the bioinformatic prioritization. From the method it seems that the only bioinformatic analysis is a simple adjustment for outliers, RIN, and gene-specific information. Is that what the bioinformatic prioritization in step 2a is referring to? I list these two items as major concerns because it is truly fundamental to how genes are ultimately prioritized, so any change dramatically changes the downstream output. For that reason, clarity is critical.

(2) It is unclear why a machine learning algorithm was applied so late in the process given that no replication of the analysis could be completed, and overfitting thus remains a major concern. There were only 44 genes in the machine learning step, does this analysis really add much beyond a simple univariate model with correction given that no downstream validation of the impressive model performance is provided? Perhaps the AMP-AD datasets could be used for validation of the proposed model?

(3) More detail on the neuron marker analysis within the bulk RNA sequencing dataset that was mentioned to be excluded needs to be detailed in this manuscript. SERPINA5 protein appears to be entirely absent in the control brains that are stained, and then co-localizes with tangle forming neurons in disease, leaving open the possibility that this is a repair process that is specific to neurons rather than tangles. Recognizing that the two are quite challenging to differentiate, it would still be helpful to better understand the degree to which SERPINA5 RNA expression is correlated with the proportion of neurons in these homogenate samples, particularly to better

understand how that may change over the course of disease (if it does). Currently, it is hard to know what SERPINA5 would be doing in the brain, so any additional insight into cell-specific processes is critical for interpretation.

Minor Concerns

(4) The monotonic assumption is curious, and in fact the SERPINA5 staining results (with a slight drop in ghost tangles) illustrates why this may not be the ideal assumption. What was the reason for assuming a monotonic increase? Was it simply a proxy for tangle burden given the categorical grouping, and did it actually end up adding much in terms of reducing the search space? It seems that in discovery having a wider net would be preferable given the planned replication (as the process expansion demonstrated), but I also recognize there is not much that can be done about that at this point given the NanoString runs have been completed already. For others who are planning to perform this type of analysis in the future, it still may be helpful to comment on the utility of this assumption.

(5) It is quite notable that the top results from the extreme phenotype analysis all came from the process networks. Could the authors clarify this a bit? It sounds like DE genes were annotated to enriched pathways, the top pathways were selected, and then all the genes in those top pathways were included as candidates. Is that correct? Then step 4 would retain only those that showed an association with pathology?

(6) Did the authors note any sex differences in SERPINA5 associations? There are number of reasons to expect there might be, so it would be interesting to check.

(7) It might also be worth noting that a similar SERPINA5 pattern was observed in the temporal cortex in the AMP-AD datasets where higher expression was observed in cases compared to controls ([https://agora.ampadportal.org/genes/\(genes-router:gene-details/ENSG00000188488\)](https://agora.ampadportal.org/genes/(genes-router:gene-details/ENSG00000188488))).

(8) The authors don't provide much speculation about what could be driving this association, but I think the possibility of a non-causal role should be discussed. It is quite possible, given SERPINA5's well established biological roles mentioned by the authors, that this upregulation and co-localization reflects a disease response that is a consequence, rather than a cause, of disease.

Signed by Timothy J. Hohman, PhD, Associate Professor of Neurology, Vanderbilt University Medical Center

Reviewer #3 (Remarks to the Author):

This manuscript presents a novel multidisciplinary approach that combines RNA-seq and Nanostring validation, rational bioinformatic methods, digital pathology, machine learning, and advanced biostatistics to uncover potential mediators of hippocampal vulnerability during the progression of AD. A truly unique and powerful aspect of the study is leveraging the availability of hippocampi samples from clinically and pathologically well-characterized cases of not only typical AD, but also cases of cortical and hippocampal-dominant atypical AD (a field in which the senior author and co-authors at Mayo Jax have considerable expertise). In this study, comparing these AD subtypes allowed for the identification, classification and prioritization of relevant genes dysregulated in a monotonic manner such that directionality of expression change increased or decreased as control > cortical-dominant > typical > hippocampal-dominant in order to home in on mechanisms of hippocampal neuronal vulnerability. However, another novelty of the study is that these diagnostic groups could also be leveraged to discover subtype-specific alterations.

In any event, the authors eventually identify five genes that meet the prioritization criteria of their multilayered approach: Serpina5, Rybp, Slc38a2, Fem1b, and Pydc1. Of these, the serine protease inhibitor SERPINA5 was validated as co-labeling with early and late tau pathology and potentially interacting with tau via co-IP studies. Hence, SERPINA5 may be involved in hippocampal

vulnerability in AD via its interactions with tau and future directions will explore this possibility

This a well-presented and rigorous study. Its biggest strength, aside from the identification of SERPINA5-mediated activity as a new pathway to study in AD pathogenesis, is captured in the authors statement that, "In this era of big data generation, scientists need advanced methods to prioritize biologically relevant targets from large, heterogenous human datasets." No argument here and this approach could be broadly adapted to other questions in AD and other neurodegenerative diseases as a whole, since selective neuronal vulnerability is a hallmark phenomenon.

The two limitations of the study are that: 1) the MMSEs of the subjects show that they are late-stage cases and, as such, the identified gene changes could be epiphenomenal and not involved in pathogenesis per se; and 2) the co-IP experiments need to be better explained or expanded upon to be more quantitative since it is unclear how many cases were used and how many different controls were applied. The proposition that SERPINA5 binds and interacts with tau – as opposed to simply being another molecule trapped in the NFT – is a major strength, so these aspects of the methods should be more rigorously described and/or presented. A final note is that it might be nice to provide a brief description of the other four genes in the Discussion. For instance, SLC38A2 is involved in BBB function and RYBP is a Polycomb protein with potential epigenetic consequences.

On the whole, however, this study is a tour de force from an outstanding team with great impact potential for driving future AD research.

Dear Reviewers,

Thank you for the opportunity to resubmit our manuscript and especially for the time taken to review our responses. Please find inline structured responses to reviewer critiques. Beneath each reviewer's critique, we provide a **response to the reviewer in blue font**, which includes relevant line numbers from the text. **Quoted text is provided in green font** with specific information where the text was incorporated. In the markup copy of the text, we have additionally commented to which Reviewer critique (Rc) the edits were intended.

Best,

Melissa E. Murray

REVIEWER COMMENTS

Reviewer #1 (Remarks to the Author):

In this study, Crist et al. integrated multiple approaches to identify disease-relevant genes that associated with neuropathologically diagnosed Alzheimer's disease (AD). They specifically show that SERPINA5 is a novel Tau-binding partner in the hippocampus that potentially regulates the maturation of neurofibrillary tangles. The authors first conducted bulk transcriptome profiling in hippocampi from a cohort comprising 40 AD cases and 15 controls to identify AD-associated genes and validated their findings in an expanded NanoString cohort comprising 150 AD cases and 32 controls. The authors integrated bulk RNA sequencing profiling with quantitative Tau and amyloid-beta pathology to uncover genes related to the typical AD phenotype and extreme phenotypes. Furthermore, the authors adopted a machine learning method to identify 5 top candidate genes as neuropathologically diagnosed disease predictors in the human brain. They subsequently showed that one of the top genes, encodes SERPINA5, is a Tau-binding partner.

Overall, this work is straightforward. The topic is significant and of general interest to a broader audience, which is within the scope of Nature Communications. However, in the current study, both the analytical and experimental evidence are not sufficient to substantiate their conclusion. Some crucial analytical and experimental details are missing and the rationale of the analytical approaches is not clear.

Specifically:

1. The authors did not provide details on their RNAseq analysis, e.g. whether they consider the age and gender effects of individuals.

We appreciate this comment and have worked to make this clearer throughout the text. We purposely did not adjust for covariates such as age, sex and genetics early on in our gene selection process because the AD subtypes exhibit a constellation of clinicopathologic features. These AD subtypes are defined by an algorithm that classifies patterns of thioflavin-S-positive neurofibrillary tangles in the hippocampus versus cortical regions (Murray et al. *Lancet Neurology* 2011), but these subtypes exhibit their own unique patterns regarding age and sex (Murray et al. *Lancet Neurology* 2011, Janocko et al. *Acta Neuropath* 2012, Hanna Al-Shaikh et al. *JAMA Neurology* 2019). For example, hippocampal sparing AD cases are overrepresented in men, have a younger at the age of onset, and faster cognitive decline. In comparison, limbic predominant AD cases are overrepresented in women, have an older age of onset, and slower cognitive decline. Adjusting for age and sex early on in our process may have negated the meaningfulness of these biological variables in using the subtypes (i.e. hippocampal vulnerability). To address covariates, we have provided a spider/radar plot showing each of our top 5 genes and its association with APOE status, sex, age at death, and cellular markers (Figure 4h). We have added text to provide clarity in our introduction (line 83), results (line 107) and methods sections (line 570).

Introduction Text: Further examination of their clinical relevance revealed a constellation of clinical and demographic differences that were monotonically-directed among AD subtypes, including striking differences in age at onset, sex, and APOE $\epsilon 4$ status¹⁰⁻¹³.

Results Text: Given the constellation of clinicopathologic features found to differ among AD subtypes, we did not adjust RNA-Seq data by age, sex, or APOE ε4 status.

Extended Data Methods Text: A generalized linear model, adjusting for RIN, was applied with the underlying distribution set as a negative binomial to obtain log2FC and p-value of gene expression differences between the above-mentioned phenotypic groups. Given the biological relevance of age, sex, and APOE ε4 status to AD subtypes¹⁰⁻¹³, these covariates were not used to adjust RNA-Seq data.

- Furthermore, the selection criteria of the differentially expressed genes associated with AD subtypes are not stringent enough. Specifically, when using edgeR to determine the significantly differentially expressed genes, the authors used a filtering FDR < 0.25. As the commonly used cutoffs of filtering criteria are < 0.1, is there a reference to support this filtering criterion?

Thank you for the opportunity to clarify. As the reviewer recommends, the current convention in the field of bioinformatics for FDR cutoff of differentially expressed genes uses an FDR < 0.1. In our study, the selection of an FDR of < 0.25 (rather than < 0.10) was to enable a broad set of genes to be chosen. Our plan was to prioritize genes using a less stringent criterion to avoid excluding any relevant genes. We would like to refer the reviewer to a quote from the Broad Institute: “An FDR of 25% indicates that the result is likely to be valid 3 out of 4 times, which is reasonable in the setting of exploratory discovery where one is interested in finding candidate hypothesis to be further validated as a results of future research. Given the lack of coherence in most expression datasets and the relatively small number of gene sets being analyzed, using a more stringent FDR cutoff may lead you to overlook potentially significant results.” Taken from: https://www.gsea-msigdb.org/gsea/doc/GSEAUUserGuideFrame.html?Interpreting_GSEA and discussed in Bonnet BMC Genomics 2013 (<https://bmcgenomics.biomedcentral.com/articles/10.1186/1471-2164-14-904>) Therefore, we opted to stay broad in the beginning then to perform validation in a larger NanoString cohort and explore biological relevance after narrowing down using neuropathology.

We have performed the exercise of applying an FDR of < 0.10 at the beginning. The table below reflects the differences in using a 0.25 versus 0.10 FDR:

Criteria	Representative Phenotype # genes	Extreme Phenotype # genes
FDR < 0.25 only	2727	107
FDR < 0.25 LogFC < -1 and LogFC > 1	141	73
FDR < 0.10 only	851	35
FDR < 0.10 LogFC < -1 and LogFC > 1	112	29

If we would have used an FDR of < 0.10, we would have lost 2 genes that met inclusion criteria for NanoString analysis: *OR7A5* and *DYDC2*. We have updated the text to clarify our use of FDR < 0.25 instead of FDR < 0.10 (line 581).

Extended Data Methods Text: To enable a broad set of differentially expressed genes to be selected for downstream prioritization efforts utilizing neuropathology measures, bioinformatic prioritization employed a less stringent cutoff (FDR<0.25) than current convention (i.e., FDR<0.10). To further enrich for genes found to be downregulated or upregulated by a factor of 2 or higher, log2FC >1 or <-1 cutoff was applied.

- Also, there is a lack of details on the approaches of identifying the differentially expressed genes associated with AD phenotypes (Fig. 2; Step 1 and 3). In particular, the authors did not include the rationale of how the 31 genes were selected based on literatures (Fig. 2; Step 1), and the details of how the cellular processes and the corresponding genes were identified (Fig. 2; Step 3).

To clarify these topics we have updated our Results section and elaborated in the methods section to describe how the literature based genes were selected (line 101 and line 554).

Results Text: We selected 25 genes located at loci identified in genome-wide association studies (GWAS) of late-onset AD¹⁴⁻¹⁹, 3 genes associated with early-onset AD^{20,21}, and 3 additional genes identified to modify the phenotype of AD²²⁻²⁴ (**Fig. 2 [Step 1]**).

Extended Data Methods Text: Known genes associated with clinical expression of AD were selected in (**Fig. 2 [Step 1]**). From large consortium efforts, we evaluated 25 genes that are located at loci identified in GWAS of late-onset AD: *ABCA7*, *APOE*, *BIN1*, *CD2AP*, *CD33*, *CLU*, *CR1*, *EPHA1*, *MS4A4E*, *MS4A6A*, *PICALM*, *CASS4*, *CELF1*, *DSG2*, *FERMT2*, *HLA-DRB1*, *HLA-DRB5*, *INPP5D*, *MEF2C*, *NME8*, *PTK2B*, *RIN3*, *SLC24A4*, *SORL1*, *ZCWPW1*¹⁴⁻¹⁹. The remaining literature-based genes included those associated with early-onset AD (*APP*, *PSEN1*, *PSEN2*)^{20,21} or those implicated in modifying the phenotype of AD (*GRN*, *SIRT1*, *TOMM40*)²²⁻²⁴.

For the cellular process, we used a patented MetaCore algorithm designed by Clarivate Analytics. This algorithm was designed by Clarivate Analytics scientists who pre-built the process networks with an excerpt from the company website included below:

*“The **pre-built networks** (Process Networks, Disease Biomarker Networks, Metabolic Networks, etc.) are reviewed by Clarivate Analytics’ scientists. These Networks focus on building specific and complete interconnected networks focused on topics in the respective category. There will be objects that appear in multiple networks in multiple categories.”* Taken from: https://support.clarivate.com/LifeSciences/s/article/MetaCore-Data-source-for-MetaCore-pathway-maps-networks-and-general-ontologies?language=en_US

We have emphasized that the network was pre-built in the Results section and updated this in the methods section (line 128 and line 588).

Results Text: Thus, **our third step** was to identify differentially expressed genes from Step 2a that were enriched in pre-built process networks using a curated database created by MetaCore™ to model main cellular processes (**Fig. 2 [Step 3]**)

Extended Data Methods Text: Significantly perturbed networks were identified by mapping differentially expressed genes from each phenotype onto pre-built process networks.

- Also, in the current study, an independent validation cohort was not used to validate the performance of the prediction model. A separate dataset for validation should be included.

We agree in principle that statistical models require external validation. The purpose of our prediction model, and the research of the paper, was not to develop and validate the prediction model. The prediction model was a means to identify and prioritize genes for further study. As a result, the study design was built to validate using the cost-effective NanoString technology and to provide biological validation. To biologically validate our findings, the top 5 genes were examined for their impact on global neuropathology and cognitive measures in Figure 4e-g.

A validation study for the prediction model would represent a new series of (expensive) experiments and would not be a feasible step forward without extensive grant resources and tissue. Given the validation of the model was not, nor was it intended to be, a focus of the paper, we provide additional data that was available to help validate the prioritization of the approach. In particular, to address reviewer's concern we now provide differential expression analysis from three AMP-AD validation sets to illustrate the generalizability of the top 5 genes. We have incorporated the results from *SERPINA5* into the Results (line 250) and Discussion (line 367), added pertinent information to the Methods (line 704) and Extended Data Fig. 17 (line 1452), and elaborate in the figure legend of Extended Results 2 (line 1506).

Results text: Examination of three validation datasets from AMP-AD^{25,38,39} (**Extended Results 2**) revealed *SERPINA5* to be significantly upregulated in AD compared to controls in the temporal cortex (Mayo-TCX: FDR=0.000040, log₂FC=2.1), superior temporal gyrus (Mount Sinai brain bank [MSBB]-Brodmann area [BM]22: FDR=0.095, log₂FC=1.0), and parahippocampal gyrus (MSBB-BM36: FDR=0.0016, log₂FC=2.0).

Discussion text: In addition, *SERPINA5* expression was significantly upregulated in AD in all three AMP-AD validation datasets from temporal cortex and parahippocampal gyrus^{25,38,39}.

Methods text: **AMP-AD RNA-Seq Validation Datasets** We downloaded gene read counts and metadata files of three RNA-Seq cohorts from AD knowledge portal on Synapse (www.synapse.org, **Extended Data Fig. 17a**), each consisting of postmortem brains from AD and controls (**Extended Data Fig. 17b**). These included the Mayo Clinic temporal cortex (Mayo-TCX)^{25,38}, and Mount Sinai VA Medical Center Brain Bank superior temporal gyrus (MSBB-BM22) and parahippocampal gyrus (MSBB-BM36)³⁹. The gene read counts were generated through a consensus bioinformatics pipeline that aligned raw RNA-Seq data files, counted reads mapped to genes, and reported quality control measures. Further details can be found on the AD knowledge portal (<https://adknowledgeportal.synapse.org>, Synapse ID: syn17010685) and by Wan et al.⁹⁵. Neuropathologic information provided in the available metadata files was used to assign individuals as AD, control, or other (**Extended Data Fig. 17b**). AD cases and controls were the focus of our differential expression analyses. As previously described in the "Gene differential expression and enrichment analysis for RNA-Seq data" section, we performed differential expression analysis between AD and control samples using R edgeR package⁸⁹ and cqn package⁸⁷. The covariates included in the design matrix were diagnosis (AD or control, categorical), RIN (continuous), age at death (continuous), sex (categorical) and flowcell (categorical) for all three datasets, plus the source of samples (categorical) for Mayo-TCX dataset.

Extended Data Fig. Extended Data Figure 17 | AMP-AD RNA-Seq validation datasets reprocessed from Mayo Clinic and Mount Sinai brain bank. a, Dataset names correspond to those referenced throughout the manuscript. DoD is provided, except where data was generated by study authors (NET) and shared within the AMP-AD knowledge portal. SynapseID's can be searched directly within the portal only (<https://adknowledgeportal.synapse.org/>). b, The AMP-AD knowledge portal was accessed to download raw gene count and associated metadata files, which was subsequently inspected and underwent quality control. aThe total number of unique samples IDs that contained the gene count file downloads for a given brain region and dataset prior to exclusions. Sample IDs were excluded if RNA-Seq gene counts had inconsistent values or did not have an associated metadata file; if sex was inconsistent between inference of Y chromosome gene expression and sex noted in their associated metadata files; if RIN <5; if gene counts (counts per million) from principal components analysis of gene expression identified outliers (PC1 or PC2 >4SD from mean); if recommended by Mayo-TCX parent study (syn6126114), and for MSBB if rRNA >5% or the sample with the lowest number of reads in a unique individual that contained two sets of RNA-Seq data (duplicates), and if ethnographic status was indicated as other than non-Hispanic white to match the current study. hNeuropathologically diagnosed AD samples from the Mayo RNA-Seq dataset had a Braak tangle stage ≥IV and controls had a Braak tangle stage ≤III^{25,38}. iNeuropathologically diagnosed AD samples from the MSBB RNA-Seq dataset had a Braak tangle stage ≥IV and CERAD neuropathologic category ≥2, whereas controls had a Braak tangle stage ≤III and CERAD neuropathology category ≤139. jAny samples lacking metadata needed for analysis, had associated neuropathologic information not matching prescribed diagnostic criteria, or samples that contained a non-AD or non-control neuropathologic diagnosis were excluded from our analyses and classified as "other/unknown". c, Age at death, proportion of females, and frequency of the *APOE* ε4 risk allele are provided for each AMP-AD dataset. Data are presented as: sample size (percentage) or median (25th percentile, 75th percentile). Acronyms: DoD=Date of download,

MSBB=Mount Sinai VA Medical Center Brain Bank, n/a=not applicable, RNA-Seq=RNA sequencing, TCX=temporal cortex.

Extended Results 2 legend: To further examine the relevance of the top 5 genes outside the hippocampus, we investigated three AMP-AD validation datasets. We present differential expression analysis of *SERPINA5*, *RYBP*, *SLC38A2*, *FEM1B*, and *PYDC1* in the temporal cortex (Mayo-TCX syn3163039^{25,38}, MSBB-BM22 syn20801188³⁹) and the parahippocampal gyrus (MSBB-BM36 syn20801188³⁹). *SERPINA5* and *SLC38A2* were significantly upregulated across all three validation datasets. *RYBP* was significantly upregulated in two of the three validation datasets. *FEM1B* performed in the opposite direction compared to the current study. *PYDC1* was significantly downregulated across all three validation datasets.

Mayo-TCX: Mayo Clinic Temporal Cortex (syn3163039, TCX)

GeneName	log2FC	FDR	p-value	meanCQN.control	meanCQN.AD	AvgMappedReads
SERPINA5	2.1	0.000040	0.00000085	-0.15	1.5	103
RYBP	0.0049	1.0	0.95	5.6	5.7	3165
SLC38A2	1.0	0.0000015	0.0000000086	6.9	7.7	9841
FEM1B	-0.10	0.24	0.072	6.3	6.3	5092
PYDC1	-0.78	0.00077	0.000039	3.2	3.2	77

MSBB-BM22: Mt. Sinai Brain Bank Superior Temporal Cortex (syn20801188, BM22)

GeneName	log2FC	FDR	p-value	meanCQN.control	meanCQN.AD	AvgMappedReads
SERPINA5	1.0	0.095	0.00040	-0.57	0.23	21
RYBP	0.14	0.34	0.029	4.3	4.4	1024
SLC38A2	0.35	0.27	0.015	6.3	6.6	4105
FEM1B	-0.038	0.82	0.40	6.0	6.0	3173
PYDC1	-0.27	0.59	0.15	1.2	0.89	17

MSBB-BM36: Mt. Sinai Brain Bank Superior Temporal Cortex (syn20801188, BM36)

GeneName	log2FC	FDR	p-value	meanCQN.control	meanCQN.AD	AvgMappedReads
SERPINA5	2.0	0.0016	0.0000030	-1.1	0.81	31
RYBP	0.17	0.051	0.0015	4.5	4.6	1077
SLC38A2	0.29	0.28	0.039	6.3	6.6	3710
FEM1B	-0.10	0.21	0.022	6.2	6.0	3180
PYDC1	-0.46	0.066	0.0024	2.2	1.7	22

Crist et al. (Mayo Clinic Hippocampus)

GeneName	log2FC	FDR	p-value	Mean.Ctrl	Mean.Typical	AvgMappedReads
SERPINA5	1.5	0.0062	0.000012	-0.87	0.68	220
RYBP	0.25	0.12	0.0074	3.2	3.4	3659
SLC38A2	0.93	0.0025	0.0000022	4.7	5.5	13845
FEM1B	0.19	0.14	0.010	4.5	4.7	9333
PYDC1	-1.3	0.043	0.00056	0.36	-0.87	18

5. While the authors showed that *SERPINA5* expression is associated with the AD subtypes and the maturation of neurofibrillary tangles, there is insufficient evidence showing this protein is a Tau-binding partner (the interaction of *SERPINA5* and Tau was indeed barely detectable in AD brain Fig. 6H), and whether it is involved in tangle maturation.

We appreciate this comment and agree that more could be done to convince the reader of *SERPINA5*-Tau binding. We would like to clarify that the reason we see such small bands in the *SERPINA5*-Tau co-IP compared to 10% input is due to the low levels of *SERPINA5* expression in the hippocampus compared to tau (we keep the exposure settings the same for both antibodies). As *SERPINA5* is only expressed in a small proportion of total neurons and AD patients already exhibit hippocampal atrophy, when immunoprecipitating it is challenging to obtain high yields. To address the reviewer's concern, we performed additional experiments using tissue from another brain region (i.e. frontal cortex) where we can use a larger volume of tissue to obtain higher yields. We have expanded the Results (line 315) and extended methods sections (line 828), as well as the extended data legend (line 1482).

Results text: To investigate this further, we performed a co-immunoprecipitation (co-IP) experiment to determine if we detected a *SERPINA5*-tau protein-protein interaction. Bulk hippocampal tissue was taken from an AD case and control and used to immunoprecipitate *SERPINA5*, which was followed by immunoblotting using the pan-tau marker, E1⁴⁵. We observed a strong tau signal in the total homogenate (input) of both AD cases and controls, but only detected a tau band in the *SERPINA5* co-IP of the AD

case (**Fig. 6h**). This initial study provided supportive evidence that SERPINA5 acts as a tau protein binding partner. To provide stronger evidence to support our hypothesis of a SERPINA5-tau interaction, we expanded our studies to the frontal cortex where larger tissue volumes can be dissected for greater yields. Western blot analysis of co-immunoprecipitation experiment revealed that tau was successfully pulled down by SERPINA5 in all three AD cases, but not in controls, suggesting that SERPINA5-tau interaction is disease-specific (**Fig. 6h**).

Extended methods text: Tissue was sampled from frozen hippocampi of a 73 year old male control (Braak=I, Thal=0) and an 86 year old male AD case (Braak=V, Thal=5). A second set of independent cases and controls was additionally investigated. For these, tissue was sampled from frozen mid-frontal cortices from three AD cases (#1 64 year old female [Braak=VI, Thal=5]; #2 60 year old female [Braak=VI, Thal=4]; #3 68 year old female [Braak VI; Thal 4]) and three controls (#4 75 year old female [Braak=I, Thal=0]; #5 78 year old male [Braak=II, Thal=1]; #6 96 year old female [Braak=II, Thal=3]). Raw western blot images can be found in **Extended Data Fig. 19-20**.

Extended Data legend text: **Extended Data Figure 20 | Raw images of western blots from co-IP of frontal cortex.** Included are the original, uncropped images of the SERPINA5 immunoprecipitation and tau (E1 antibody) immunoblot shown from frontal cortex in **Fig. 6h** after chemiluminescence exposure for 30 seconds. Note that the input represents the total homogenate before IP (exposure time 10 seconds). Tissue was sampled from frozen frontal cortices from three AD cases (#1 64 year old female [Braak=VI, Thal=5]; #2 60 year old female [Braak=VI, Thal=4]; #3 68 year old female [Braak VI; Thal 4]) and three controls (#4 75 year old female [Braak=I, Thal=0]; #5 78 year old male [Braak=II, Thal=1]; #6 96 year old female [Braak=II, Thal=3]). Acronyms: AD=Alzheimer's Disease, Ctl=control, IP= immunoprecipitation.

Specific points

(Note to reviewer: To ensure we adequately addressed all concerns, we broke out the second paragraph of the Reviewer 1's synopsis. As a result, some of our replies to the specific points below that were addressed above may feel repetitive)

6. Regarding the differential gene expression analysis, despite providing RINs, the authors need to clarify whether they adjusted for covariates, such as age, sex, genetic effects, etc.

Please see Critique 1 outlined above.

7. When the authors distinguished AD from controls by using the prioritized genes identified in the training dataset, the validation dataset was not independent from the training dataset. A separate data set for validation should be included.

Please see Critique 4 outlined above.

8. It is important to determine whether the 5 AD genes identified are specific to AD. Thus, the predictive values of these genes for other diseases (at least for some neurodegenerative diseases) should be evaluated.

We wholeheartedly agree with this comment! Generating predictive values for these genes in other diseases would require performing RNA-Seq or NanoString studies on hippocampi extracted from other tauopathies and neurodegenerative disorders. While we do not have the capacity or funding to perform new sequencing experiments in other diseases, we do plan to assess our top gene, SERPINA5, in other tauopathies. Our goal for this manuscript is to remain focused on hippocampal vulnerability in AD by highlighting a unique and effective approach to identify genes in a biologically complex system. We feel that including other tauopathies and neurodegenerative disorders in this manuscript would take away from our main focus (i.e. hippocampal vulnerability in AD). We are glad that this reviewer brought up this important point, but hope they understand the need to separate the two studies to allow us to maintain focus.

9. The rationale and data for the selection of SERPINA5 should be clearly presented.

We appreciate this feedback. We have gone back through the text and clarified our rationale for SERPINA5 selection (line 238). For the reviewers, we have also modified our Figure 2 workflow to highlight SERPINA5's journey.

Results text: To put into context of the above-mentioned steps, *SERPINA5* was first prioritized in the differential expression analysis of the representative phenotype (Step 2a: $p=0.00012$; Step 2b: false discovery rate[FDR]=0.0062, $\log_2(\text{foldchange}[FC])=1.5$). The translational neuropathology approach (Step 4) further identified *SERPINA5* gene expression levels to be monotonically directed ($p=0.00422$, Fig. 5a) and associate robustly with neuropathologic measures (robust association: Braak tangle stage $p<0.0001$ and $FDR<0.10$; Ab39 burden $p<0.002$ and $FDR<0.22$, and Thal amyloid phase $p<0.001$ and $FDR<0.14$, Extended Data Fig. 3). Random Forest analyses (Step 5, Fig. 4a-b, Extended Data Fig. 8) identified *SERPINA5* as the 2nd top predictor in the representative phenotype (mean minimum depth=2.32, $p<0.0001$), 3rd top predictor in the extreme phenotype (mean minimum depth=2.97, $p<0.0001$), and as the top predictor in the overall discrimination of AD from control (mean minimum depth=2.18, $p<0.0001$).

10. While the top 5 predictive genes are presented for each comparison in the prediction model (Extended Data Fig. 8), it is unclear how many genes passed the threshold (i.e., "high-predictive genes") and how many "high-predictive genes" overlapped between the typical and extreme phenotypes.

Thank you for the opportunity to clarify. Our "top genes" were selected based on prioritization using a judicious application of Random Forest and consideration of feasibility to pursue biological significance. This was not based on any statistical parameters. We have updated the results (line 196) and methods (line 666) to reflect this. These top genes enabled us to form a composite measure for the purpose of validating against cognitive impairment (MMSE), global neuropathologic severity (Braak and Thal), and neuropathologic characteristics (digital pathology). A threshold was not necessarily applied to identify "high-predictive genes," but instead the in-silico experiments enabled us to rank the "importance" of the gene expression measures when discerning between two groups. Mean minimum depth was ranked and a top 5 cutoff was applied to facilitate feasibility of downstream examination. We now provide the Extended Results 1 spreadsheet and legend (line 1506), which provides an overview of importance measures.

Results text: To facilitate prioritization and feasibility of evaluating top genes, mean minimum depth was used to rank the genes and a cutoff at the top 5 was applied (Extended Results 1).

Methods text: Five top genes rather than 6 or 10 were arbitrarily chosen based on prioritization and feasibility for downstream applications, not based on statistical selection.

Extended Results 1 legend: Extended Results 1: Random Forest analyses were applied to examine the “importance” of gene expression changes when discriminating between groups. The main analysis examined differences between controls versus all AD subtypes with sub-analyses of the representative phenotype and extreme phenotype. Judicious examination of the mean minimum depth and times a root was ranked to facilitate identifying the top 5 genes.

Minor points:

11. Figure 1b: Need to ensure terminology is used consistently: “RNA-Seq cohort” and “NanoString Cohort.”
12. Figure 3a: Regarding the expression of DE Genes, “ $1 \geq \log FC \leq -1$,” the labelling is incorrect. It should be $\log FC > 1$ or < -1 .

Thank you for your attention to detail and highlighting these issues. The figures are now updated.

b

RNA-Seq cohort (n=55)

NanoString cohort (n=182)

Figure 3

Bioinformatic Prioritization: DE Genes with $FDR < 0.25$ and $1 > \log_2 FC < -1$

Reviewer #2 (Remarks to the Author):

The manuscript from Crist and colleagues leverages an extreme phenotype design and comprehensive neuropathological assessment to identify genes that may be associated with tau dysregulation in Alzheimer's disease, focusing particularly on the hippocampus. The study design is creative and makes use of incredibly detailed neuropathological characterization to carefully define each study group. The small sample size is then overcome by leveraging annotation and bioinformatic techniques to improve statistical power, a replication dataset leveraging the emerging NanoString platform, and multi-modal validation to confirm the top candidate including co-IP using microscopy. The results implicate SERPINA5 as a novel tau binding partner worth pursuing in experimental follow-up to clarify causality, while also raising 4 additional novel candidate genes relevant to tau changes in AD. Overall, this is a very strong study and the SERPINA5 results are quite compelling. Some limitations that could strengthen the manuscript are listed below.

Major Concerns

1. Given the prioritization approach taken, each step of gene selection is critical and only some steps are well described. The selection of AD loci for example includes proximal genes even when the functional gene in the region is known to be elsewhere (the inclusion of CELF1 instead of SPI1 is one example). Meanwhile many additional causal genes that are well validated are not included (ADAM10, TREM2, UNC5C, PLGC2, PILRA). The gene symbol presented in the GWAS papers is rarely a functionally validated gene, but many of the regions now do have functionally validated genes that can be leveraged in this type of analysis. The poor gene list that is included in the first step undermines the downstream processes and dramatically reduces the value of including known AD genes. There is a nice review that recently came out that provides a quality summary of the known v. unknown genes at AD loci from the GWAS papers: <https://www.sciencedirect.com/science/article/pii/S0969996120302515>.

We would like to thank the reviewer for the recommendation and especially highlighting this review. To address this, we've identified all 112 genes from the Neuner et al. 2020 review and created an excel file as Extended Results 3 that takes each gene through our prioritization method (line 403, line 561, line 1514). Of the 112 Neuner Review genes, 108 Ensemble ID's were available for comparison in our dataset. One gene (*ACE*) had two available Ensemble IDs and 4 genes were not detected in our dataset. Of these, 3 additional genes would have passed our criteria: *ADAM10*, *ANKMY2*, *ATP5F1*. The literature gene *PSEN2* is currently included in the study. We have additionally addressed these genes in the limitations section and will look forward to including these genes in our future studies and highlighting them for the greater community.

Discussion text: Furthermore, the selection of literature-based genes does not include several contemporary genes, which may dramatically reduce the value of including known genes that can be leveraged using our novel approach. To provide a contemporary examination, we applied our translational neuropathology approach from Step 4 to a recently summarized gene set from Neuner et al.⁷⁸ to identify genes that may associate with hippocampal vulnerability in AD. We identified three additional protein-coding genes that exhibited monotonic directionality among AD subtypes and robustly associated with neuropathologic measures: *ADAM10*, *ANKMY2*, *ATP5F1* (**Extended Results 3**). We encourage the research community to investigate these and other genes from our NanoString gene list in future studies.

Extended Data Methods text: A recent review by Neuner and colleagues⁷⁸, highlighted 112 genetic loci associated with AD (**Extended Results 3**). Of the 112 genes, one gene (*ACE*) had two available Ensemble IDs and 4 genes were not detected in our dataset. Of the 108 genes with Ensemble ID's available for post-hoc comparison in our RNA-Seq dataset, 3 additional genes would have passed our criteria: *ADAM10*, *ANKMY2*, *ATP5F1*. The literature gene *PSEN2* is currently included in the study.

Extended Results 3 legend text: **Extended Results 3:** A contemporary literature-based gene set derived from Neuner, et al.⁷⁸ was additionally examined using the translational neuropathology approach from Step 4. The goal was to provide an updated gene set and put into context of the current study. Three additional genes met inclusion criteria, which will be included in future studies: *ADAM10*, *ANKMY2*, *ATP5F1*.

2. A related challenge is the bioinformatic prioritization. From the method it seems that the only bioinformatic analysis is a simple adjustment for outliers, RIN, and gene-specific information. Is that what the bioinformatic prioritization in step 2a is referring to? I list these two items as major concerns because it

is truly fundamental to how genes are ultimately prioritized, so any change dramatically changes the downstream output. For that reason, clarity is critical.

We thank the reviewer for the opportunity to clarify. We did not adjust for covariates such as age, sex and genetics early on in our gene selection process as the AD subtypes exhibit a constellation of these clinicopathologic features. The AD subtypes are neuropathologically classified using an algorithm that interprets Thio S-positive neurofibrillary tangle patterns in the hippocampus and association cortices (Murray *et al.* Lancet Neurology 2011). In addition to the monotonically-directed tangle patterns in the hippocampus and cortices, we have observed monotonic patterns with regard to demographic and clinical differences (Murray *et al.* Lancet Neurology 2011, Janocko *et al.* Acta Neuropath 2012, Murray-Cannon *et al.* Acta Neuropath 2014, Hanna Al-Shaikh *et al.* JAMA Neurology 2019):

- Female sex (HpSp < Typical < Limbic predominant)
- *APOE* ϵ 4 (HpSp < Typical < Limbic predominant)
- *MAPT* H1H1 (HpSp < Typical < Limbic predominant)
- Age onset (HpSp < Typical < Limbic predominant)
- MMSE final score (HpSp < Typical < Limbic predominant)
- Age at death (HpSp < Typical < Limbic predominant)

Adjusting for these variables early on in our process may have negated the meaningfulness of these biological variables in using the subtypes (i.e. hippocampal vulnerability). To address covariates, we have provided a spider/radar plot showing each of our top 5 genes and its association with *APOE* status, sex, age at death, and cellular markers (Figure 4h). We have added text to provide clarity in our introduction (line 83), results (line 107) and methods sections (line 570).

Introduction text: Further examination of their clinical relevance revealed a constellation of clinical and demographic differences that were monotonically-directed among AD subtypes, including striking differences in age at onset, sex, and *APOE* ϵ 4 status¹⁰⁻¹³.

Results Text: Given the constellation of clinicopathologic features found to differ among AD subtypes, we did not adjust RNA-Seq data by age, sex, or *APOE* ϵ 4 status.

Extended Data Methods Text: A generalized linear model, adjusting for RIN, was applied with the underlying distribution set as a negative binomial to obtain log₂FC and p-value of gene expression differences between the above-mentioned phenotypic groups. Given the biological relevance of age, sex, and *APOE* ϵ 4 status to AD subtypes¹⁰⁻¹³, these covariates were not used to adjust RNA-Seq data.

3. It is unclear why a machine learning algorithm was applied so late in the process given that no replication of the analysis could be completed, and overfitting thus remains a major concern. There were only 44 genes in the machine learning step, does this analysis really add much beyond a simple univariate model with correction given that no downstream validation of the impressive model performance is provided? Perhaps the AMP-AD datasets could be used for validation of the proposed model?

We employed a diverse team-science based approach used to discover relevant gene expression changes. We sought to employ an early prioritization strategy that relied upon phenotypic differences more than dichotomization of AD versus Control. New ideas were brought in when additional team members joined. The team is interested in going back and reevaluating decisions made earlier in the process, but this represents future research that will need to be conducted. While further empirical and

methodologic work is needed to understand if the machine learning approach is superior to the univariate approach, it is intuitively considered to be far superior in this case. It is widely appreciated that ensemble of "weak learner" (many small simple trees joined together to form a single prediction) is a superior method of generating predictions. The simple models minimize the overfitting that the reviewer raises rightfully as a concern. From a biological process, there is no expectation that genes work independently and that using models that will allow for synergy of the gene signatures is biologically relevant.

We agree in principle that statistical models require external validation. The purpose of our prediction model, and the research of the paper, was not to develop and validate the prediction model. The prediction model was a means to identify and prioritize genes for further study. As a result, the study design was built to validate using the cost-effective NanoString technology and to provide biological validation. To biologically validate our findings, the top 5 genes were examined for their impact on global neuropathology and cognitive measures in Figure 4e-g.

A validation study for the prediction model would represent a new series of (expensive) experiments and would not be a feasible step forward without extensive grant resources and tissue. Given the validation of the model was not, nor was it intended to be, a focus of the paper, we provide additional data that the reviewer recommended to help validate the prioritization of the approach. In particular, to address reviewer's concern we now provide differential expression analysis from three AMP-AD validation sets to illustrate the generalizability of the top 5 genes. We have incorporated the results from *SERPINA5* into the Results (line 250) and Discussion (line 367), added pertinent information to the Methods (line 704) and Extended Data Fig. Z (line 1452), and elaborate in the figure legend of Extended Results 2 (line 1506).

Results text: Examination of three validation datasets from AMP-AD^{25,38,39} (**Extended Results 2**) revealed *SERPINA5* to be significantly upregulated in AD compared to controls in the temporal cortex (Mayo-TCX: FDR=0.000040, log₂FC=2.1), superior temporal gyrus (Mount Sinai brain bank [MSBB]-Brodmann area [BM]22: FDR=0.095, log₂FC=1.0), and parahippocampal gyrus (MSBB-BM36: FDR=0.0016, log₂FC=2.0).

Discussion text: In addition, *SERPINA5* expression was significantly upregulated in AD in all three AMP-AD validation datasets from temporal cortex and parahippocampal gyrus^{25,38,39}.

Methods text: **AMP-AD RNA-Seq Validation Datasets** We downloaded gene read counts and metadata files of three RNA-Seq cohorts from AD knowledge portal on Synapse (www.synapse.org, **Extended Data Fig. 17a**), each consisting of postmortem brains from AD and controls (**Extended Data Fig. 17b**). These included the Mayo Clinic temporal cortex (Mayo-TCX)^{25,38}, and Mount Sinai VA Medical Center Brain Bank superior temporal gyrus (MSBB-BM22) and parahippocampal gyrus (MSBB-BM36)³⁹. The gene read counts were generated through a consensus bioinformatics pipeline that aligned raw RNA-Seq data files, counted reads mapped to genes, and reported quality control measures. Further details

can be found on the AD knowledge portal (<https://adknowledgeportal.synapse.org>, Synapse ID: syn17010685) and by Wan et al.⁹⁵. Neuropathologic information provided in the available metadata files was used to assign individuals as AD, control, or other (**Extended Data Fig. 17b**). AD cases and controls were the focus of our differential expression analyses. As previously described in the “Gene differential expression and enrichment analysis for RNA-Seq data” section, we performed differential expression analysis between AD and control samples using R edgeR package⁸⁹ and cqn package⁸⁷. The covariates included in the design matrix were diagnosis (AD or control, categorical), RIN (continuous), age at death (continuous), sex (categorical) and flowcell (categorical) for all three datasets, plus the source of samples (categorical) for Mayo-TCX dataset.

Extended Data Fig. 17 text: Extended Data Figure 17 | AMP-AD RNA-Seq validation datasets reprocessed from Mayo Clinic and Mount Sinai brain bank. a, Dataset names correspond to those referenced throughout the manuscript. DoD is provided, except where data was generated by study authors (NET) and shared within the AMP-AD knowledge portal. SynapseID's can be searched directly within the portal only (<https://adknowledgeportal.synapse.org/>). **b**, The AMP-AD knowledge portal was accessed to download raw gene count and associated metadata files, which was subsequently inspected and underwent quality control. ^aThe total number of unique samples IDs that contained the gene count file downloads for a given brain region and dataset prior to exclusions. Sample IDs were excluded ^bif RNA-Seq gene counts had inconsistent values or did not have an associated metadata file; ^cif sex was inconsistent between inference of Y chromosome gene expression and sex noted in their associated metadata files; ^dif RIN <5; ^eif gene counts (counts per million) from principal components analysis of gene expression identified outliers (PC1 or PC2 >4SD from mean); ^fif recommended by Mayo-TCX parent study (syn6126114), and for MSBB if rRNA >5% or the sample with the lowest number of reads in a unique individual that contained two sets of RNA-Seq data (duplicates), and ^gif ethnographic status was indicated as other than non-Hispanic white to match the current study. ^hNeuropathologically diagnosed AD samples from the Mayo RNA-Seq dataset had a Braak tangle stage ≥IV and controls had a Braak tangle stage ≤III^{25,38}. ⁱNeuropathologically diagnosed AD samples from the MSBB RNA-Seq dataset had a Braak tangle stage ≥IV and CERAD neuropathologic category ≥2, whereas controls had a Braak tangle stage ≤III and CERAD neuropathology category ≤1³⁹. ^jAny samples lacking metadata needed for analysis, had associated neuropathologic information not matching prescribed diagnostic criteria, or samples that contained a non-AD or non-control neuropathologic diagnosis were excluded from our analyses and classified as “other/unknown”. **c**, Age at death, proportion of females, and frequency of the *APOE* ε4 risk allele are provided for each AMP-AD dataset. Data are presented as: sample size (percentage) or median (25th percentile, 75th percentile). Acronyms: DoD=Date of download, MSBB=Mount Sinai VA Medical Center Brain Bank, n/a=not applicable, RNA-Seq=RNA sequencing, TCX=temporal cortex.

Extended Results 2 legend text: Extended Results 2: To further examine the relevance of the top 5 genes outside the hippocampus, we investigated three AMP-AD validation datasets. We present differential expression analysis of *SERPINA5*, *RYBP*, *SLC38A2*, *FEM1B*, and *PYDC1* in the temporal cortex (Mayo-TCX syn3163039^{25,38}, MSBB-BM22 syn20801188³⁹) and the parahippocampal gyrus (MSBB-BM36 syn20801188³⁹). *SERPINA5* and *SLC38A2* were significantly upregulated across all three validation datasets. *RYBP* was significantly upregulated in two of the three validation datasets. *FEM1B* performed in the opposite direction compared to the current study. *PYDC1* was significantly downregulated across all three validation datasets.

Mayo-TCX: Mayo Clinic Temporal Cortex (syn3163039, TCX)

GeneName	log2FC	FDR	p-value	meanCQN.control	meanCQN.AD	AvgMappedReads
SERPINA5	2.1	0.000040	0.00000085	-0.15	1.5	103
RYBP	0.0049	1.0	0.95	5.6	5.7	3165
SLC38A2	1.0	0.0000015	0.0000000086	6.9	7.7	9841
FEM1B	-0.10	0.24	0.072	6.3	6.3	5092
PYDC1	-0.78	0.00077	0.000039	3.2	3.2	77

MSBB-BM22: Mt. Sinai Brain Bank Superior Temporal Cortex (syn20801188, BM22)

GeneName	log2FC	FDR	p-value	meanCQN.control	meanCQN.AD	AvgMappedReads
SERPINA5	1.0	0.095	0.00040	-0.57	0.23	21
RYBP	0.14	0.34	0.029	4.3	4.4	1024
SLC38A2	0.35	0.27	0.015	6.3	6.6	4105
FEM1B	-0.038	0.82	0.40	6.0	6.0	3173
PYDC1	-0.27	0.59	0.15	1.2	0.89	17

MSBB-BM36: Mt. Sinai Brain Bank Superior Temporal Cortex (syn20801188, BM36)

GeneName	log2FC	FDR	p-value	meanCQN.control	meanCQN.AD	AvgMappedReads
SERPINA5	2.0	0.0016	0.0000030	-1.1	0.81	31
RYBP	0.17	0.051	0.0015	4.5	4.6	1077
SLC38A2	0.29	0.28	0.039	6.3	6.6	3710
FEM1B	-0.10	0.21	0.022	6.2	6.0	3180
PYDC1	-0.46	0.066	0.0024	2.2	1.7	22

Crist et al. (Mayo Clinic Hippocampus)

GeneName	log2FC	FDR	p-value	Mean.Ctrl	Mean.Typical	AvgMappedReads
SERPINA5	1.5	0.0062	0.000012	-0.87	0.68	220
RYBP	0.25	0.12	0.0074	3.2	3.4	3659
SLC38A2	0.93	0.0025	0.0000022	4.7	5.5	13845
FEM1B	0.19	0.14	0.010	4.5	4.7	9333
PYDC1	-1.3	0.043	0.00056	0.36	-0.87	18

4. More detail on the neuron marker analysis within the bulk RNA sequencing dataset that was mentioned to be excluded needs to be detailed in this manuscript. SERPINA5 protein appears to be entirely absent in the control brains that are stained, and then co-localizes with tangle forming neurons in disease, leaving open the possibility that this is a repair process that is specific to neurons rather than tangles. Recognizing that the two are quite challenging to differentiate, it would still be helpful to better understand the degree to which SERPINA5 RNA expression is correlated with the proportion of neurons in these homogenate samples, particularly to better understand how that may change over the course of disease (if it does). Currently, it is hard to know what SERPINA5 would be doing in the brain, so any additional insight into cell-specific processes is critical for interpretation.

We completely understand this concern. We did not originally include the neuronal marker in our radar plots to avoid regressing out genes relevant to tangle bearing neurons. The reviewer has suggested an insightful way to examine the relationship, which we now graphically represent in Extended Data Fig.14 and elaborate upon in the Results and Discussion (line 271, line 381, line 1433).

Results text: Interestingly, gene expression levels of *SERPINA5* and the neuronal marker *ENO2* inversely associate in both controls ($R=-0.55$, $p=0.001$) and AD cases ($R=-0.35$, $p<0.001$) (Extended Data Fig. 14).

Discussion text: Although, *SERPINA5*-immunopositive structures were only observed in the context of neurofibrillary tangle pathology, gene expression studies revealed an inverse association between *SERPINA5* and the neuronal marker *ENO2* in hippocampi of both controls and AD. This may suggest that upregulation of *SERPINA5* represents a repair process in neurons. Currently, it is unclear if *SERPINA5* upregulation is a consequence of failed adaptation to neuronal dysfunction, or a contributor to neurofibrillary tangle formation.

Extended Data Figure text: Extended Data Figure 14 | *SERPINA5* gene expression correlates with neuronal marker *ENO2*. Gene expression levels from the neuronal marker *ENO2* associated with

SERPINA5 levels in both controls (R=-0.55, p=0.001) and AD cases (R=-0.35, p<0.001) from the NanoString cohort.

Minor Concerns

5. The monotonic assumption is curious, and in fact the SERPINA5 staining results (with a slight drop in ghost tangles) illustrates why this may not be the ideal assumption. What was the reason for assuming a monotonic increase? Was it simply a proxy for tangle burden given the categorical grouping, and did it actually end up adding much in terms of reducing the search space? It seems that in discovery having a wider net would be preferable given the planned replication (as the process expansion demonstrated), but I also recognize there is not much that can be done about that at this point given the NanoString runs have been completed already. For others who are planning to perform this type of analysis in the future, it still may be helpful to comment on the utility of this assumption.

Thank you for the thoughtful question. The monotonic directionality was purposeful in that the AD subtypes represent a constellation of clinicopathologic features. By relying upon what the biology has taught us over the past 10 years, we sought to identify gene expression changes that may mirror the phenotypic observations. By limiting to the number of tangles in the severity spectrum, we may be comparing early versus late stage cases rather than “resilient” versus “vulnerable”. The AD subtypes are neuropathologically defined by an algorithm that interprets Thio S-positive neurofibrillary tangle patterns in the hippocampus and association cortices (Murray *et al.* Lancet Neurology 2011). The monotonic nature of these AD subtypes revealed itself several times over, which prompted deeper investigation in the gene expression changes that may similarly demonstrate monotonic directionality.

Monotonically-directed clinical and demographic features:

- Female sex (HpSp < Typical < Limbic predominant)
- APOE e4 (HpSp < Typical < Limbic predominant)
- MAPT H1H1 (HpSp < Typical < Limbic predominant)
- Age onset (HpSp < Typical < Limbic predominant)
- MMSE final score (HpSp < Typical < Limbic predominant)
- Age at death (HpSp < Typical < Limbic predominant)

This consistent pattern of monotonic ordering prompted our further investigation into what may underlie these changes, by first focusing on the hippocampus. The neurofibrillary tangle images in Figure 6 were a result of identifying the top gene as a tau binding partner and was not the monotonically directed pattern used. We highlight the neurofibrillary tangle lifespan to demonstrate the dynamic nature of tau and SERPINA5. We have added text to help clarify and comment on the utility of this assumption (line 83, line 394)

Introduction text: Further examination of their clinical relevance revealed a constellation of clinical and demographic differences that were monotonically-directed among AD subtypes, including striking differences in age at onset, sex, and APOE ε4 status¹⁰⁻¹³.

Discussion text: . The applied assumption of monotonic directionality was a reflection of the biology of the AD subtypes observed when examining clinicopathologic and demographic differences. It is important to note that the utility of this assumption may have excluded genes relevant to hippocampal vulnerability in AD that did not track with AD subtypes.

6. It is quite notable that the top results from the extreme phenotype analysis all came from the process networks. Could the authors clarify this a bit? It sounds like DE genes were annotated to enriched pathways, the top pathways were selected, and then all the genes in those top pathways were included as candidates. Is that correct? Then step 4 would retain only those that showed an association with pathology?

Yes, the reviewer is correct. For the process networks, we used a patented MetaCore algorithm designed by Clarivate Analytics. This algorithm was designed by Clarivate Analytics scientists who pre-built the process networks with an excerpt from the company website included below:

*“The pre-built networks (**Process Networks**, Disease Biomarker Networks, Metabolic Networks, etc.) are reviewed by Clarivate Analytics’ scientists. These Networks focus on building specific and complete interconnected networks focused on topics in the respective category. There will be objects that appear in multiple networks in multiple categories.”* Taken from:

https://support.clarivate.com/LifeSciences/s/article/MetaCore-Data-source-for-MetaCore-pathway-maps-networks-and-general-ontologies?language=en_US

We have emphasized that the network was pre-built in the Results section and updated this in the methods section (line 128 and line 588).

Results Text: Thus, **our third step** was to identify differentially expressed genes from Step 2a that were enriched in pre-built process networks using a curated database created by MetaCore™ to model main cellular processes (**Fig. 2 [Step 3]**). All genes from the top process networks that were selected for the representative phenotype and separately for the extreme phenotype were then included as candidate genes.

Extended Data Methods Text: Significantly perturbed networks were identified by mapping differentially expressed genes from each phenotype onto pre-built process networks.

7. Did the authors note any sex differences in SERPINA5 associations? There are number of reasons to expect there might be, so it would be interesting to check.

Thank you for the opportunity to elaborate. Within the context of controls and AD subtypes, we did not observe a sex difference. To address this we now provide a statistical summary in the Results section (line 260) and visually display the findings in Extended Data Fig. 12 (line 1419).

Results text: We did not observe differences in *SERPINA5* gene expression levels when males and females were stratified within controls ($p=0.92$), hippocampal sparing AD ($p=0.61$), typical AD ($p=0.83$), or limbic predominant AD ($p=0.87$) (**Extended Data Fig. 12**).

Extended Data Fig. text: **Extended Data Figure 12 | SERPINA5 gene expression stratified by sex.** SERPINA5 gene expression levels in the NanoString cohort did not differ when males and females were stratified within controls, hippocampal sparing AD, typical AD, or limbic predominant AD. Each jitter plot overlay displays females on the left (darker color) and males on the right (lighter color)

8. It might also be worth noting that a similar SERPINA5 pattern was observed in the temporal cortex in the AMP-AD datasets where higher expression was observed in cases compared to controls ([https://agora.ampadportal.org/genes/\(genes-router:gene-details/ENSG00000188488\)](https://agora.ampadportal.org/genes/(genes-router:gene-details/ENSG00000188488))).

Thank you for this helpful recommendation! We have now run our top genes through the AMP-AD datasets and address SERPINA5 pattern in the Results (line 250) and Discussion (line 367).

Results text: Examination of three validation datasets from AMP-AD^{25,38,39} (**Extended Results 2**) revealed *SERPINA5* to be significantly upregulated in AD compared to controls in the temporal cortex (Mayo-TCX: FDR=0.000040, log₂FC=2.1), superior temporal gyrus (Mount Sinai brain bank [MSBB]-Brodmann area [BM]22: FDR=0.095, log₂FC=1.0), and parahippocampal gyrus (MSBB-BM36: FDR=0.0016, log₂FC=2.0).

Discussion text: In addition, *SERPINA5* expression was significantly upregulated in AD in all three AMP-AD validation datasets from temporal cortex and parahippocampal gyrus^{25,38,39}.

9. The authors don't provide much speculation about what could be driving this association, but I think the possibility of a non-causal role should be discussed. It is quite possible, given *SERPINA5*'s well established biological roles mentioned by the authors, that this upregulation and co-localization reflects a disease response that is a consequence, rather than a cause, of disease.

We agree that there is a possibility that *SERPINA5* is upregulated in response to the disease rather than acting as a cause. We have added a section into the discussion (line 381).

Discussion text: Although, *SERPINA5*-immunopositive structures were only observed in the context of neurofibrillary tangle pathology, gene expression studies revealed an inverse association between *SERPINA5* and the neuronal marker *ENO2* in hippocampi of both controls and AD. This may suggest that upregulation of *SERPINA5* represents a repair process in neurons. Currently, it is unclear if *SERPINA5* upregulation is a consequence of failed adaptation to neuronal dysfunction, or a contributor to neurofibrillary tangle formation. Our co-IP experiments provide supportive evidence that *SERPINA5* and tau interact in the hippocampus and cortex of AD brains. This was complemented by our microscopy experiments that suggest *SERPINA5* could be a tipping point between pretangles and mature tangles. Future studies will be directed at investigating the role of *SERPINA5* in neurons with a focus on identifying whether *SERPINA5* acts in a casual or non-causal fashion. This information could aid in identifying a mechanism to prevent or slow neuronal death.

Reviewer #3 (Remarks to the Author):

This manuscript presents a novel multidisciplinary approach that combines RNA-seq and Nanostring validation, rational bioinformatic methods, digital pathology, machine learning, and advanced biostatistics to uncover potential mediators of hippocampal vulnerability during the progression of AD. A truly unique and powerful aspect of the study is leveraging the availability of hippocampi samples from clinically and pathologically well-characterized cases of not only typical AD, but also cases of cortical and hippocampal-dominant atypical AD (a field in which the senior author and co-authors at Mayo Jax have considerable expertise). In this study, comparing these AD subtypes allowed for the identification, classification and prioritization of relevant genes dysregulated in a monotonic manner such that directionality of expression change increased or decreased as control > cortical-dominant > typical > hippocampal-dominant in order to home in on mechanisms of hippocampal neuronal vulnerability. However, another novelty of the study is that these diagnostic groups could also be leveraged to discover subtype-specific alterations.

In any event, the authors eventually identify five genes that meet the prioritization criteria of their multilayered approach: Serpina5, Rybp, Slc38a2, Fem1b, and Pydc1. Of these, the serine protease inhibitor SERPINA5 was validated as co-labeling with early and late tau pathology and potentially interacting with tau via co-IP studies. Hence, SERPINA5 may be involved in hippocampal vulnerability in AD via its interactions with tau and future directions will explore this possibility

This a well-presented and rigorous study. Its biggest strength, aside from the identification of SERPINA5-mediated activity as a new pathway to study in AD pathogenesis, is captured in the authors statement that, "In this era of big data generation, scientists need advanced methods to prioritize biologically relevant targets from large, heterogenous human datasets." No argument here and this approach could be broadly adapted to other questions in AD and other neurodegenerative diseases as a whole, since selective neuronal vulnerability is a hallmark phenomenon.

We thank you the reviewer for the thorough summary and enthusiasm for our manuscript.

The two limitations of the study are that:

1. The MMSEs of the subjects show that they are late-stage cases and, as such, the identified gene changes could be epiphenomenal and not involved in pathogenesis per se; and

Yes, this is an important point. We leverage the range of hippocampal involvement in the AD subtypes toward the effort of understanding implications of the gene changes from a mildly involved hippocampal sparing case to extremely involved limbic predominant case. Regardless, we made sure to highlight this limitation for our readers in the discussion (line 398).

Discussion text: The monotonic directionality of hippocampal involvement facilitated evaluation of gene expression changes from selective resilience in hippocampal sparing AD to selective vulnerability in limbic predominant AD. Although this range enabled deeper investigation into late-stage AD cases, more work is needed to determine the relevance of these gene expression changes as epiphenomenal versus active players in the pathogenesis of hippocampal vulnerability.

2. The co-IP experiments need to be better explained or expanded upon to be more quantitative since it is unclear how many cases were used and how many different controls were applied. The proposition that SERPINA5 binds and interacts with tau – as opposed to simply being another molecule trapped in the NFT – is a major strength, so these aspects of the methods should be more rigorously described and/or presented.

We thank the reviewer for bringing up this point and have worked to clarify how many cases and controls were investigated. We have updated this in the text (line 315, 828, 1314, 1482, 1489) and performed additional experiments using frontal lobe tissue to provide the readers more evidence of this interaction.

Results text: To investigate this further, we performed a co-immunoprecipitation (co-IP) experiment to determine if we detected a SERPINA5-tau protein-protein interaction. Bulk hippocampal tissue was taken from an AD case and control and used to immunoprecipitate SERPINA5, which was followed by immunoblotting using the pan-tau marker, E1⁴⁵. We observed a strong tau signal in the total homogenate (input) of both AD cases and controls, but only detected a tau band in the SERPINA5 co-IP of the AD case (**Fig. 6h**). This initial study provided supportive evidence that SERPINA5 acts as a tau protein binding partner. To provide stronger evidence to support our hypothesis of a SERPINA5-tau interaction,

we expanded our studies to the frontal cortex where larger tissue volumes can be dissected for greater yields. Western blot analysis of co-immunoprecipitation experiment revealed that tau was successfully pulled down by SERPINA5 in all three AD cases, but not in controls, suggesting that SERPINA5-tau interaction is disease-specific (**Fig. 6h**).

Extended methods text: Tissue was sampled from frozen hippocampi of a 73 year old male control (Braak=I, Thal=0) and an 86 year old male AD case (Braak=V, Thal=5). A second set of independent cases and controls was additionally investigated. For these, tissue was sampled from frozen mid-frontal cortices from three AD cases (#1 64 year old female [Braak=VI, Thal=5]; #2 60 year old female [Braak=VI, Thal=4]; #3 68 year old female [Braak VI; Thal 4]) and three controls (#4 75 year old female [Braak=I, Thal=0]; #5 78 year old male [Braak=II, Thal=1]; #6 96 year old female [Braak=II, Thal=3]). Raw western blot images can be found in **Extended Data Fig. 19-20**.

Figure legend 6h text: (Left) Tissue was sampled from frozen hippocampi of a 73 year old male control (Braak=I, Thal=0) and an 86 year old male AD case (Braak=V and Thal=5). (Right) Tissue was sampled from frozen frontal cortices of a 75 year old female control (Braak=I, Thal=0) and a 68 year old male AD case (Braak=VI and Thal=5). Note that uncropped Western blots can be found in **Extended Data Fig. 19-20**.

Figure legend Extended Data Figure 19: Tissue was sampled from frozen hippocampi of a 73 year old male control (Braak=I, Thal=0) and an 86 year old male AD case (Braak=V, Thal=5).

Figure legend Extended Data Figure 20: Extended Data Figure 20 | Raw images of western blots from co-IP of frontal cortex. Included are the original, uncropped images of the SERPINA5 immunoprecipitation and tau (E1 antibody) immunoblot shown from frontal cortex in **Fig. 6h** after chemiluminescence exposure for 30 seconds. Note that the input represents the total homogenate before IP (exposure time 10 seconds). Tissue was sampled from frozen frontal cortices from three AD cases (#1 64 year old female [Braak=VI, Thal=5]; #2 60 year old female [Braak=VI, Thal=4]; #3 68 year old female [Braak VI; Thal 4]) and three controls (#4 75 year old female [Braak=I, Thal=0]; #5 78 year old male [Braak=II, Thal=1]; #6 96 year old female [Braak=II, Thal=3]). Acronyms: AD=Alzheimer's Disease, Ctl=control, IP= immunoprecipitation.

3. A final note is that it might be nice to provide a brief description of the other four genes in the Discussion. For instance, SLC38A2 is involved in BBB function and RYBP is a Polycomb protein with potential epigenetic consequences.

We agree that this would help provide a well-balanced discussion and have added the recommended paragraph that briefly describes the function of our top 5 genes (line 344).

Discussion text: Upon reviewing the published literature, our top 5 proteins were found to have a multitude of diverse functions within the living system. In addition to SERPINA5's canonical serine protease inhibition function, it exhibits broad protease activity and interacts with phospholipids⁴⁹⁻⁵³, glycosaminoglycans⁵⁴, proteins⁵⁵⁻⁵⁸, and cysteine proteases⁵⁹. These properties make SERPINA5 extremely unique among serine protease inhibitors. RYBP is a multifunctional protein that binds several transcriptional factors⁶⁰, mediates histone H2A monoubiquitination⁶¹ and plays roles in development^{62,63}, apoptosis⁶⁴⁻⁶⁶ and cancer⁶⁰. SLC38A2 functions as a sodium-dependent amino acid transporter that mediates both the efflux of neutral α -amino acids across the blood-brain barrier and their uptake into neurons⁶⁷⁻⁶⁹. FEM1B is an ankyrin repeat protein that induces replication stress signaling through its interaction with checkpoint kinase 1⁷⁰, regulates ubiquitin-protein transferase activity⁷¹ and promotes apoptosis⁷². PYDC1 (also known as ASC2/POP1) regulates the innate immune response by suppressing NF κ B transcription factor activity and pro-caspase-1 activation⁷³.

On the whole, however, this study is a tour de force from an outstanding team with great impact potential for driving future AD research.

Thank you for taking the time to review our paper and we appreciate the vote of confidence!

REVIEWER COMMENTS

Reviewer #1 (Remarks to the Author):

The authors have included justifications for not conducting the covariate adjustment in the revised study. However the justifications are not solid enough. For instance, in Figure 4H, one of the top candidates, RYBP, is obviously associated with the age at death rather than tangle or Abeta (yellow colour). Since the sample size of the discovery dataset is small, the authors need to carefully interpret the results.

Moreover, in this revised version, the authors used additional AD brain transcriptome data to verify their findings. In the Method section for "AMP-AD RNA-Seq Validation Datasets", the authors explicitly stated that the RIN, age at death, sex, and flow cell information were included as covariates when analysing these datasets. Thus, it is suggested that the same analysis methods be applied to the datasets used for identification of the candidate markers at the discovery stage. Meanwhile, the sample size of the individual datasets used for the replication should be stated in the Method section.

Reviewer #2 (Remarks to the Author):

The authors have adequately addressed all of my comments and suggestions. The additional clarifications have improved the manuscript, and the remaining concerns that could not be fully addressed are trivial. This is an important manuscript that contributes critical new knowledge to the field.

Reviewer #3 (Remarks to the Author):

The authors have satisfactorily addressed the few mild/moderate concerns of the initial submission. My view is that the first two reviews were also addressed thoroughly. It was understood that there were demographic and practical rationale for relaxing controls for age and gender, as well as FDR in order to capture more candidates for validation. The inclusion of FC in addition to HIP for tau co-IPs allayed concerns that the interaction might not be that functionally compelling for disease progression in these subtypes. It will be very interesting for the field to see if follow-up external validation studies support the present hypotheses.

REVIEWER COMMENTS

Reviewer #1 (Remarks to the Author):

The authors have included justifications for not conducting the covariate adjustment in the revised study. However the justifications are not solid enough. For instance, in Figure 4H, one of the top candidates, RYBP, is obviously associated with the age at death rather than tangle or Abeta (yellow colour). Since the sample size of the discovery dataset is small, the authors need to carefully interpret the results.

Moreover, in this revised version, the authors used additional AD brain transcriptome data to verify their findings. In the Method section for "AMP-AD RNA-Seq Validation Datasets", the authors explicitly stated that the RIN, age at death, sex, and flow cell information were included as covariates when analysing these datasets. Thus, it is suggested that the same analysis methods be applied to the datasets used for identification of the candidate markers at the discovery stage. Meanwhile, the sample size of the individual datasets used for the replication should be stated in the Method section.

Thank you for the time you have taken to review the resubmission. While adjustment by age and sex is commonplace, we must continue to respectfully disagree with the need to adjust for these variables at the discovery stage in a study with defined biological phenotypes. Our approach is unique in utilizing Alzheimer's disease (AD) subtypes that were neuropathologically defined. Past studies have identified that age and sex differences are inherent to the AD subtypes. We hypothesize that age and sex may be important aspects of their biology and should not be used to adjust as it may diminish gene expression changes that track with selective vulnerability.

Identification of a strong relationship of age with *RYBP* could be an important finding to understanding hippocampal vulnerability that we hope to follow-up on in future discovery studies. We would like to find a balance with how our novel approach may impact the field and respect the critique provided. To this end, we have updated the table in Extended Results 2 that contains the AMP-AD validation dataset findings of our top 5 genes. We provide a comparison of differential expression analyses of AMP-AD findings to the current study. The log2FC and significance are improved by adjustment of age, sex, RIN, and flowcell, which highlights our lenient approach that does not adjust remains valid in post-hoc analysis of the data.

Unadjusted differential expression analysis of top 5 genes

GeneName	log2FC	FDR	p-value	Mean.Ctrl	Mean.Typical
SERPINA5	1.5	0.0062	0.000012	-0.87	0.68
RYBP	0.25	0.12	0.0074	3.2	3.4
SLC38A2	0.93	0.0025	0.0000022	4.7	5.5
FEM1B	0.19	0.14	0.010	4.5	4.7
PYDC1	-1.3	0.043	0.00056	0.36	-0.87

Differential expression analysis of top 5 genes adjusted by age, sex, RIN, and flowcell

GeneName	log2FC	FDR	p-value	Mean.Ctrl	Mean.Typical
SERPINA5	1.7	0.0061	0.0000041	-0.87	0.68
RYBP	0.28	0.12	0.0027	3.2	3.4
SLC38A2	0.93	0.0050	0.0000026	4.7	5.5
FEM1B	0.20	0.17	0.0062	4.5	4.7
PYDC1	-1.0	0.21	0.0089	0.36	-0.87

To highlight the specificity of not adjusting to our approach of investigating distinct clinicopathologic phenotypes, we have elaborated in the limitation section to ensure the reader is made aware.

Discussion text: At the discovery stage, age and sex were not used to adjust as the AD subtypes exhibit a constellation of demographic and clinicopathologic features. We hypothesize that adjustment of these biological variables may limit identification of important gene expression changes that track with selective hippocampal vulnerability observed in the AD subtypes. As adjustment for age and sex is commonplace in bioinformatic prioritization, future studies lacking distinct clinicopathologic phenotypes should consider adjustment as appropriate to their study.

Thank you for recommending sample size incorporation into Methods section to provide the reader clarity.

Methods text: The resulting sample sizes for Mayo-TCX was n=68 controls and n=80 AD, MSBB-BM22 was n=33 controls and n=70 AD, MSBB-BM36 was n=30 controls and n=56 AD, and the current study was n=15 controls and n=20 typical AD (**Extended Data Fig. 17c, Extended Results 2**).

Reviewer #2 (Remarks to the Author):

The authors have adequately addressed all of my comments and suggestions. The additional clarifications have improved the manuscript, and the remaining concerns that could not be fully addressed are trivial. This is an important manuscript that contributes critical new knowledge to the field.

Thank you for your kind words and encouragement.

Reviewer #3 (Remarks to the Author):

The authors have satisfactorily addressed the few mild/moderate concerns of the initial submission. My view is that the first two reviews were also addressed thoroughly. It was understood that there were demographic and practical rationale for relaxing controls for age and gender, as well as FDR in order to capture more candidates for validation.

The inclusion of FC in addition to HIP for tau co-IPs allayed concerns that the interaction might not be that functionally compelling for disease progression in these subtypes. It will be very interesting for the field to see if follow-up external validation studies support the present hypotheses.

Thank you for your thorough review of our resubmission and reviewer responses.

REVIEWERS' COMMENTS

Reviewer #1 (Remarks to the Author):

The authors have addressed my comments. I recommend this manuscript for publication.